# Hijacking of nucleotide biosynthesis and deamidation-mediated glycolysis by an oncogenic herpesvirus

Quanyuan Wan [1], Leah Tavakoli[1], Ting-Yu Wang[2,3], Andrew J. Tucker[1], Ruiting Zhou[1], Qizhi Liu[2,6], Shu Feng [2,7], Dongwon Choi[4], Zhiheng He[5], Michaela U. Gack [1] & Jun Zhao [1] ✉

Kaposi's sarcoma-associated herpesvirus (KSHV) is the causative agent of Kaposi's sarcoma (KS) and multiple types of B cell malignancies. Emerging evidence demonstrates that KSHV reprograms host-cell central carbon metabolic pathways, which contributes to viral persistence and tumorigenesis. However, the mechanisms underlying KSHV-mediated metabolic reprogramming remain poorly understood. Carbamoyl-phosphate synthetase 2, aspartate transcarbamoylase, and dihydroorotase (CAD) is a key enzyme of the de novo pyrimidine synthesis, and was recently identified to deamidate the NF-κB subunit RelA to promote aerobic glycolysis and cell proliferation. Here we report that KSHV infection exploits CAD for nucleotide synthesis and glycolysis. Mechanistically, KSHV vCyclin binds to and hijacks cyclin-dependent kinase CDK6 to phosphorylate Ser-1900 on CAD, thereby activating CAD-mediated pyrimidine synthesis and RelA-deamidation-mediated glycolytic reprogramming. Correspondingly, genetic depletion or pharmacological inhibition of CDK6 and CAD potently impeded KSHV lytic replication and thwarted tumorigenesis of primary effusion lymphoma (PEL) cells in vitro and in vivo. Altogether, our work defines a viral metabolic reprogramming mechanism underpinning KSHV oncogenesis, which may spur the development of new strategies to treat KSHV-associated malignancies and other diseases.

Kaposi's sarcoma-associated herpesvirus (KSHV) causes at least three human malignancies or lymphoproliferative disorders, including Kaposi's sarcoma (KS), primary effusion lymphoma (PEL), and multicentric Castleman's disease (MCD) in AIDS patients and immunocompromised individuals[1–4]. More recently, KSHV was identified to cause a new syndrome, KSHV inflammatory cytokine syndrome (KICS),

which has mortality rates of up to 60%[5]. Currently, treatments for KSHV-associated malignancies are limited to the classical chemotherapy regimens typically applied to virus-negative cancers. However, clinical reports have shown that virus-associated lymphomas quickly develop drug resistance, resulting in very poor prognoses and low survival rates (e.g., 6-month median overall survival for PEL[6]). On the

[1]Florida Research and Innovation Center, Cleveland Clinic, Port St. Lucie, FL, USA. [2]Section of Infection and Immunity, Herman Ostrow School of Dentistry, University of Southern California, Los Angeles, CA, USA. [3]Proteome Exploration Laboratory, Beckman Institute, California Institute of Technology, Pasadena, CA, USA. [4]Department of Surgery, Keck School of Medicine, University of Southern California, Los Angeles, CA, USA. [5]Department of Molecular Microbiology and Immunology, Keck School of Medicine, University of Southern California, Los Angeles, CA, USA. [6]Present address: State Laboratory of Developmental Biology of Freshwater Fish, Hunan Normal University, Changsha, Hunan, China. [7]Present address: Department of Diabetes & Cancer Metabolism, Beckman Research Institute of City of Hope, Duarte, CA, USA. ✉e-mail: zhaoj6@ccf.org

other hand, the development of targeted therapies for KSHV-associated diseases has been impeded by our limited understanding of the molecular factors and processes that drive KSHV pathogenesis and tumorigenesis. As such, it is of utmost importance to investigate the detailed virus-host interactions underlying cell transformation, tumor progression and drug resistance to discover 'vulnerabilities' or molecular targets of KSHV-associated malignancies.

As a herpesvirus, KSHV exhibits a biphasic viral life cycle including a quiescent latent infection and an 'active' lytic replication[7]. It has been shown that oral epithelial cells support KSHV lytic replication[8,9], which may enable viral oral transmission[10] and latency establishment in multiple types of cells including endothelial cells and B cells. Notably, KSHV can intermittently reactivate in these cells, promoting viral persistent infections and ultimately tumorigenesis[11]. As an obligated intracellular parasite, KSHV relies on many host-cell processes and machineries for virion production and many other steps of its life cycle. Among the many host processes manipulated by KSHV are the reprogramed cellular metabolic pathways, including glycolysis[12-15], glutaminolysis[16-18], and fatty acid synthesis[13,19,20], which help facilitate virion production, cell proliferation, and tumorigenesis. Nevertheless, the mechanisms underlying viral manipulations of cellular metabolic pathways, particularly nucleotide metabolism, remain not clear[21]. Importantly, emerging studies and clinical trials have discovered that targeting altered metabolic pathways not only directly thwarts cancer growth, but also overcomes drug resistance and promotes the activity of tumor-killing T cells[22-24]. These findings collectively urge for a deepened understanding of metabolic reprogramming by KSHV to uncover unique metabolic vulnerabilities of KSHV-associated malignancies.

Glutamine amidotransferases (GATs) are a family of metabolic enzymes that utilize γ-nitrogen from glutamine for metabolic biosynthesis, including nucleotides, amino acids, glycoproteins and an enzyme cofactor NAD[25]. Our previous studies uncovered a non-canonical activity of selected GATs, which extracts the ammonia group from the glutamine and asparagine residues of proteins, resulting in a less understood protein post-translational modification (PTM): protein deamidation[26-29]. Serving as protein deamidases, GATs regulate innate immune activation[26,30], demonstrating their non-metabolic functions to control signal transduction. As one of the mammalian GATs, Carbamoyl-phosphate synthetase 2, aspartate transcarbamoylase and dihydroorotase (CAD) is a multi-functional enzyme catalyzing the first three steps of de novo pyrimidine synthesis[31]. Recently, we reported that CAD serves as a protein deamidase that targets RelA (p65), the core transcriptionally active subunit of NF-κB controlling pro-inflammatory responses and many other cell physiological activities[32,33]. Remarkably, CAD-mediated RelA deamidation at Asn-64 and Asn-139 diminishes NF-κB activation, while it promotes aerobic glycolysis via inducing the expression of key glycolytic enzymes to fuel cell proliferation[34]. Thus, the metabolic and non-metabolic activities of CAD underscore its role as a master regulator of inflammation and cellular metabolic pathways including nucleotide synthesis, glycolysis, and potentially other related pathways.

As the rate-limiting enzyme of de novo pyrimidine synthesis, the activity of CAD is allosterically regulated by uridine triphosphate (UTP)[35,36] and phosphoribosyl pyrophosphate (PRPP)[37], as well as intricately controlled by PTMs. In response to nutrient stimulation, CAD is phosphorylated and activated by MAP kinase[38] and S6K[39,40] to promote pyrimidine synthesis. During cell cycle progression, pyrimidine synthesis peaks during the S phase to accompany the high demand of nucleotides for DNA replication[41]. However, whether a cell cycle-dependent PTM modulating CAD activity is in place to support cell proliferation remains unknown. In this study, we uncover a cell cycle kinase-mediated phosphorylation that drives CAD activation by KSHV. Specifically, CAD phosphorylation is mediated by the viral homologue of human Cyclin D2, vCyclin, in concerted action with

cellular Cyclin-dependent kinase 6 (CDK6). Our data shows that vCyclin-CDK6-mediated CAD phosphorylation drives pyrimidine synthesis and RelA deamidation-mediated glycolysis to support KSHV viral persistence. Genetic depletion or pharmacological inhibition of CDK6 and CAD potently impeded KSHV lytic replication and tumorigenesis in vitro and in vivo. As such, our study uncovers a viral mechanism for reprogramming host nucleotide synthesis and glycolysis, and identifies CDK6 and CAD as druggable metabolic targets in KSHV-associated malignancies.

## Results

### KSHV infection promotes CAD activity for metabolic reprogramming

The "Warburg effect" or aerobic glycolysis describes how cells preferentially convert glucose to lactate in the presence of oxygen. Aerobic glycolysis offers a rapid ATP production, drives macromolecule biosynthesis and thus is important for fast proliferating cells such as tumor cells[42]. An increased glucose utilization has been observed in KSHV-infected cells[20], while the virus also elevates glutamine intake for nucleotide synthesis[17]. Importantly, our recent study unveiled a role for the de novo pyrimidine synthesis enzyme CAD in deamidating RelA to regulate glycolysis via upregulating enzymes and regulators in the glycolytic pathway including hexokinase 4 (HK4), pyruvate dehydrogenase kinase 2 and 3 (PDK2/PDK3), and pyruvate carboxylase (PC), a mechanism coupling nucleotide synthesis with energy metabolism[34] (Fig. 1A). We therefore hypothesized that KSHV hijacks CAD to promote de novo pyrimidine synthesis and deamidation-dependent glycolytic reprogramming, a proposed mechanism that fits the central carbon metabolic reprogramming observed during KSHV infection. To test this hypothesis, we performed targeted metabolomics analysis on KSHV-infected human oral keratinocytes HOK16B[9] (HOKs) and observed a substantial increase in the level of intracellular glycolysis intermediates (glucose, phosphoenolpyruvate (PEP), 2-phosphoglyceric acid/3-phosphoglyceric acid (2-PG/3-PG), glucose-6-phosphate/fructose-6-phosphate (G-6-P/F-6-P), fructose-1,6-biphosphate (F-1,6-BP), 1,3-biphosphoglyceric acid (1,3-BPG), dihydroxyacetone phosphate (DHAP), and glyceraldehyde-3-phosphate (GADP)), as well as CAD reaction product dihydroorotic acid and its immediate downstream pyrimidine intermediate orotic acid in KSHV-infected cells (Fig. 1B). Two-dimensional gel electrophoresis (2DGE), a system that is well established to monitor alterations in protein charge, revealed migration of RelA towards positive anode during KSHV lytic replication in HOKs and primary lymphatic endothelial cells (LECs), as well as latent infection in Tert-immortalized microvascular endothelial (TIME) cells, (Fig. 1C and S1A). Of note, this charge-dependent migration was not observed in cells reconstituted with a deamidation-resistant RelA mutant N64A[34] (Fig. S1B). Together, these results indicate that KSHV de novo infection induces RelA deamidation. Real-time qPCR analysis and immunoblotting showed that KSHV infection induced the expression, albeit at varying levels among cell types, of the previously identified RelA-deamidation-dependent metabolic genes[34] (Fig. 1A), including HK4, PC, and PDK3 (Fig. 1D–G), which was consistent with an increased lactate production in the infected HOKs, LECs and TIME cells (Fig. 1H and I). By Seahorse assay, we further observed that KSHV infection in HOKs induced aerobic glycolysis (Figs. 1J, S1C, and S1D). These data collectively indicate enhanced metabolic and deamidase enzymatic activities of CAD during KSHV infection, which correlate with the upregulation of pyrimidine synthesis and glycolysis. Of note, CAD mRNA levels showed only a marginal increase by KSHV infection (Fig. S1E), and its protein abundance remained comparable in infected HOKs and TIME cells compared to mock-infected controls (Fig. 1F and G), indicating that the enhanced activity of CAD likely stems from changes in regulatory PTMs. In support of this hypothesis, CAD purified from KSHV-infected cells exhibited greater deamidase activity than that isolated from

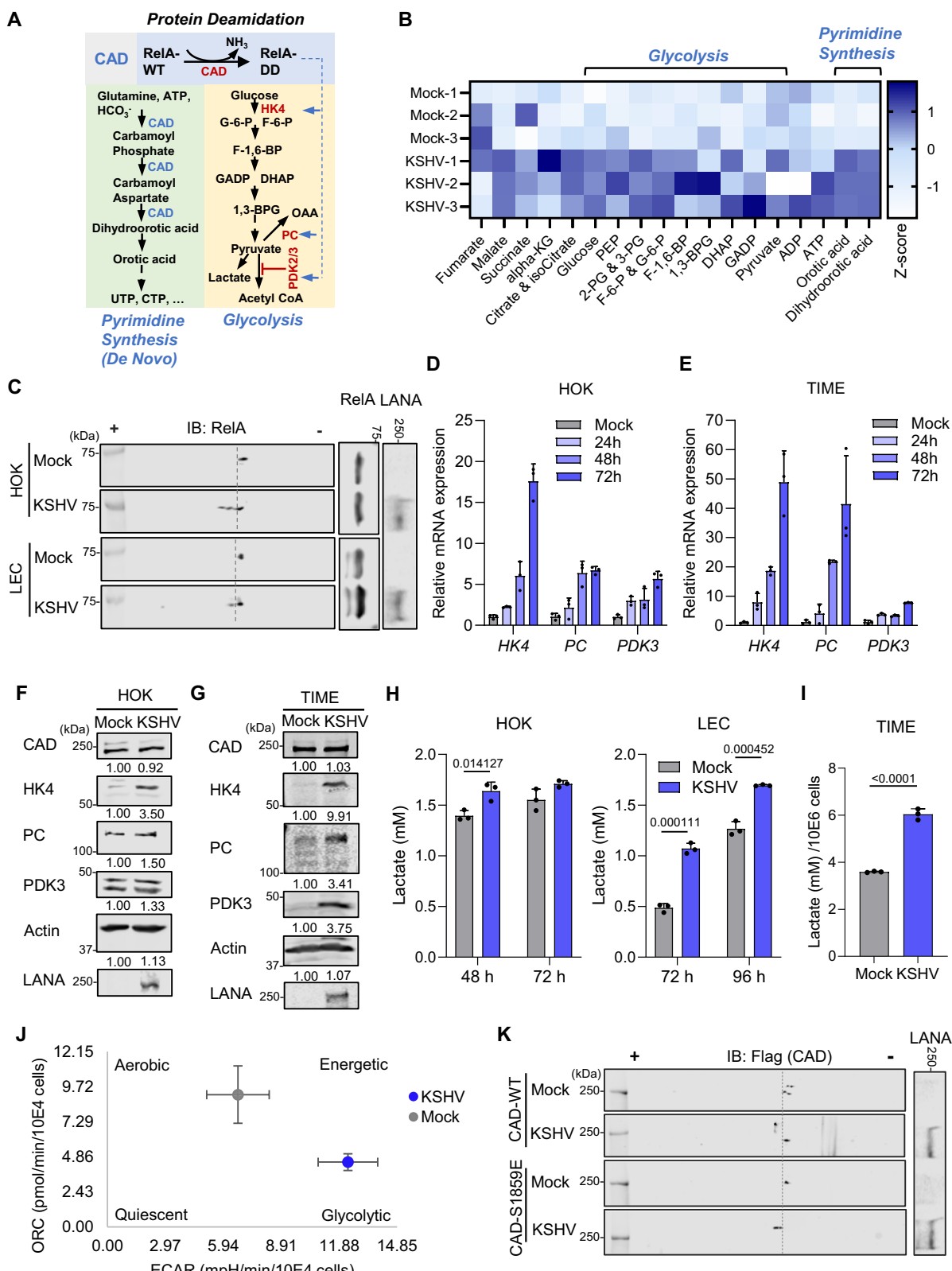

mock-infected cells at an equivalent molar concentration (Fig. S1F). CAD activity is known to be promoted by mTORC1-S6K via phosphorylation on Ser-1859[39,40]. Furthermore, KSHV infection is known to enhance mTORC1 activity[43]. Indeed, we observed an increase of CAD Ser-1859 phosphorylation in KSHV-infected cells (Fig. S1G). However, mutating Ser-1859 to glutamate (E) failed to negate the charge changes of CAD induced by KSHV infection (Fig. 1K), suggesting that KSHV

induces a different PTM on CAD which may contribute to the regulation of its enzymatic activity.

## KSHV vCyclin interacts with and promotes CAD activity

Next, we investigated the KSHV protein(s) that may interact with CAD and induce the aforementioned PTM. In co-immunoprecipitation (co-IP) assays where CAD and several latent KSHV proteins were

**Fig. 1 | KSHV infection promotes CAD activity and induces RelA deamidation.**
**A** A diagram depicting CAD in pyrimidine synthesis and in RelA deamidation to drive aerobic glycolysis via upregulating glycolysis-associating enzymes and regulators. **B** A heatmap of intracellular metabolites with z-score normalization for Human Oral Keratinocytes-16B (HOKs) infected with KSHV (MOI = 30, 48 h). Full names of the metabolites are provided in the methods. **C** HOKs and Lymphatic Endothelial Cells (LECs) were infected with KSHV (MOI = 30 and 10, respectively). Whole cell lysates (WCLs) were prepared at 24 h and analyzed by two-dimensional gel electrophoresis (2DGE) and immunoblotting. HOKs (**D**) and Tert-immortalized Microvascular Endothelial (TIME) cells (**E**) were infected with KSHV (MOI = 30 and 3, respectively). Real-time quantitative PCR (RT-qPCR) analyses of the indicated mRNAs were then performed. HOKs (**F**) and TIME cells (**G**) were infected with KSHV (MOI = 30 and 3, respectively) for 72 h. WCLs were processed in parallel and analyzed by immunoblotting with the indicated antibodies. **H** HOKs and LECs were infected with KSHV (MOI = 30 and 10, respectively) for the indicated hours. The culturing medium was collected to determine lactate concentration. **I** TIME cells were infected with KSHV (MOI = 3) for 72 h. The culturing medium was collected to determine lactate concentration at 16 h post medium replacement. **J** HOKs were infected with KSHV (MOI = 30) for 24 h. Cells were then analyzed by Seahorse assay and oxygen consumption rate (OCR) was plotted against the extracellular acidification rate (ECAR). **K** 293 T cells were transfected with plasmids expressing flag-tagged wild-type (WT) CAD or CAD phospho-mimetic mutant S1859E, and then infected with KSHV (MOI = 5) for 24 h. WCLs were analyzed by 2DGE and immunoblotting. Data are presented as mean ± SD of n = 3 biological replicates (1D, 1E, 1H, and 1I) and mean ± SEM of n = 3 biological replicates (1 J). Blots were representative of at least two independent experiments (1 C, 1 F, 1 G, and 1 K). Significance was calculated using two-tailed, unpaired Student's t-test. Source data are provided as a source data file. See also Fig. S1.

co-expressed (i.e. vIRF3, vCyclin, vFLIP, K12 (Kaposin), and LANA), we found that specifically vCyclin interacted with CAD but none of the other proteins tested (Figs. 2A, S2A, and B). Furthermore, virally expressed vCyclin co-precipitated with endogenous CAD in KSHV-infected 293 T cells and a PEL cell line BC-3 harboring KSHV (Figs. 2B and S2C). Mapping co-IP experiments showed that the glutaminase domain (GLN) and the dihydroorotase domain (DHO) of CAD were sufficient to interact with vCyclin (Fig. 2C and D), whereas the carbamoyl-phosphate synthetase domain (CPSab) showed a weaker interaction.

To assess whether vCyclin promotes CAD activity, we monitored the charge status of RelA by 2DGE and found that ectopic expression of vCyclin was sufficient to drive RelA deamidation in HOKs and 293 T cells (Figs. 2E and S2D), as well as to induce the aforementioned charge change of CAD (Fig. 2F). In vitro RelA deamidation assay showed that CAD purified from vCyclin co-expressed cells exhibited more potent deamidase activity (Fig. 2G). In agreement with the reported role for CAD-mediated RelA deamidation in NF-κB inhibition[34], we observed a reduction of RelA-mediated or Sendai virus-induced NF-κB activation in the presence of vCyclin or its cellular partner kinase, CDK6 (Figs. 2H and S2E). Moreover, we found that vCyclin induced the expression of *HK4* and *PDK3* in RelA-WT reconstituted HOKs but not in RelA-64A (deamidation-resistant) reconstituted cells (Fig. S2F), suggesting that RelA deamidation is required for vCyclin to regulate glycolytic gene expression. Targeted metabolomics analysis demonstrated that vCyclin expression in HOKs increased the level of intracellular lactate by 4-fold, indicating that vCyclin indeed promoted glycolysis (Fig. 2I). Importantly, the vCyclin-mediated upregulation of lactate concentration in the culture media was abrogated upon CAD depletion (Fig. 2J), demonstrating that vCyclin acts via CAD for glycolysis reprogramming. Finally, HOKs expressing vCyclin showed an upregulated glycolysis by Seahorse XF glycolytic rate assay (Fig. 2K). In summary, the results identified that KSHV vCyclin interacts with CAD and triggers its activation, leading to RelA deamidation and deamidation-mediated glycolysis.

## vCyclin drives pyrimidine synthesis and glycolysis during KSHV infection

To further probe the role of vCyclin in mediating metabolic reprogramming in the context of KSHV infection, we utilized a mutant recombinant BAC.16 KSHV carrying a premature stop codon in vCyclin (ΔCyclin)[44], and validated the lack of vCyclin expression in the infected cells by immunoblotting (Fig. S3A). In HOK and LEC cells, lytic replication of the corresponding BAC.16 KSHV-wild-type (WT) triggered robust RelA deamidation, which was largely abrogated in ΔCyclin-infected cells (Figs. 3A and S3B). Similarly, deamidation of RelA was abolished in ΔCyclin latently infected TIME cells (Fig. S3C). In accord, the induction of deamidation-dependent metabolic gene expression, including *HK4*, *PC*, *PDK2*, and *PDK3*, was diminished in ΔCyclin-infected HOKs, LECs, and TIME cells (Fig. S3D–G). The expression of the

metabolic genes was fully restored by overexpression of vCyclin in HOKs (Fig. S3H). In ΔCyclin-infected HOKs and TIME cells, there were significantly lower steady-state levels of metabolites within glycolysis and its branching pathway pentose phosphate pathway (PPP) (Figs. 3B and S3I). Moreover, we also observed decreased levels of pyrimidine and purine intermediates such as uridine monophosphate (UMP) and inosine monophosphate (IMP) in ΔCyclin-infected HOKs, but not for metabolites in NAD metabolism or TCA cycle (Fig. 3B). To further determine the metabolic activities of glycolysis and de novo pyrimidine synthesis, we determined the intracellular labeled metabolites after feeding the infected HOKs with [U-13C] glucose and [Amide-15N] glutamine (Fig. 3C and E). WT KSHV significantly elevated the levels of 13C-incorporated glycolysis intermediates (F-1,6-BP, DHAP, 3-PG, and Lactate) and pyrimidine intermediate UMP as compared to the mock control, while such upregulation was significantly impaired in ΔCyclin-infected HOKs (Figs. 3D and S3J). Furthermore, we observed a drastic difference in 15N-incorporated pyrimidine intermediates (dihydroorotic acid, orotic acid, and cytidine triphosphate) between WT and ΔCyclin-infected cells (Fig. 3F). Finally, when we compared the lytic replication of WT and ΔCyclin KSHV in LECs, we observed a > 95% decrease in viral mRNA expression for ΔCyclin as compared to WT KSHV (Fig. 3G). Similarly, ΔCyclin viral transcript levels were significantly reduced in HOKs compared to WT virus (Figs. 3H and S3K). Addition of uridine, a metabolite that can be converted to nucleotides via the salvage pathway, partially restored the replication of ΔCyclin without boosting viral mRNA expression (Figs. 3H, I, and S3K). RelA-DD reconstitution combined with uridine addition further increased the viral titer of ΔCyclin but failed to reach to the level of WT (Fig. S3L), whereas vCyclin overexpression fully restored the lytic replication of ΔCyclin (Fig. S3M). These data collectively indicate that KSHV vCyclin promotes glycolysis and de novo pyrimidine synthesis, and is required for optimal viral replication.

## vCyclin recruits CDK6 to phosphorylate CAD at Ser-1900

As a mammalian Cyclin D2 homologue, KSHV vCyclin hijacks cellular CDK6 to phosphorylate retinoblastoma (Rb) protein[45,46], thereby allowing cell cycle progression. We thus hypothesized that vCyclin may recruit CDK6 to phosphorylate CAD, leading to its activation in KSHV-infected cells. Indeed, CDK6 and CAD readily interacted with each other in co-immunoprecipitation assays upon ectopic overexpression (Fig. 4A), and the binding of CDK6 to endogenous CAD was enhanced in the presence of vCyclin (Fig. 4B). Furthermore, analogously to vCyclin, ectopic expression of CDK6 induced the charge change of CAD (Fig. 4C) and promoted RelA deamidation (Fig. S4A). In accord, silencing of CDK6 blocked the RelA deamidation induced by KSHV infection (Fig. S4B). Finally, by in vitro kinase assay with [γ-32P] ATP, we found that purified CDK6 potently phosphorylated CAD, which could be blocked by the specific CDK4/6 kinase inhibitor Palbociclib (Fig. 4D). These data indicated that vCyclin cooperates with CDK6 to phosphorylate CAD.

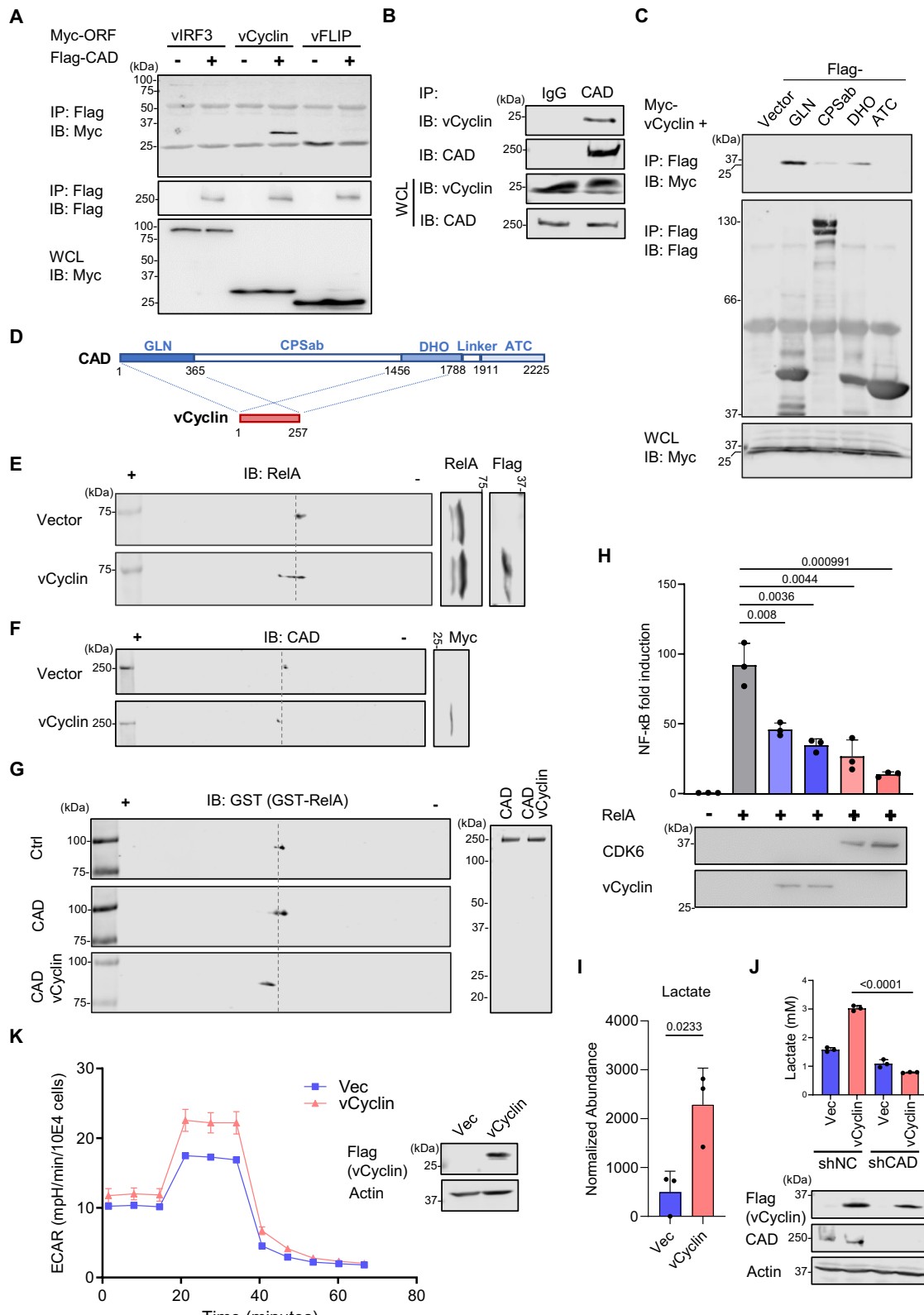

To determine the functional relevance of CDK6 in regulating CAD, we depleted endogenous CDK6 in HOKs by using two shRNAs targeting different regions (Fig. S4C). CDK6 knockdown abrogated the expression of deamidation-regulated metabolic genes *HK4, PC,* and *PDK3,* and further, reduced lytic viral gene expression (Figs. S4D and E), supporting that CDK6 is necessary for KSHV to exploit CAD-mediated RelA deamidation.

To identify the residues of CAD phosphorylated by CDK6, we performed a phospho-proteome analysis on cells that ectopically expressed vCyclin or CDK6 by tandem mass spectrometry. This analysis identified one site, Ser-1900 (S1900), whose phosphorylation was dramatically increased following vCyclin or CDK6 expression (Fig. 4E). We generated an antibody recognizing phosphorylated S1900 (p-S1900) of CAD. This phospho-specific antibody detected basal

**Fig. 2 | KSHV vCyclin interacts with CAD and drives RelA deamidation.**
**A** 293 T cells were transfected with plasmids expressing Flag-CAD and Myc-tagged KSHV latent proteins. WCLs were precipitated with anti-Flag. Precipitated proteins and WCLs were analyzed by immunoblotting with the indicated antibodies.
**B** 293 T cells were infected with KSHV (MOI = 5) for 48 h. WCLs were precipitated with IgG or anti-CAD. Precipitated proteins and WCLs were analyzed by immunoblotting with the indicated antibodies. **C, D** 293 T cells were transfected with plasmids expressing vCyclin and Flag-tagged CAD truncation mutants. WCLs were precipitated with anti-Flag. Precipitated proteins and WCLs were analyzed by immunoblotting with the indicated antibodies (**C**) and the results were summarized in (**D**). **E** HOKs were infected with lentivirus containing Flag-tagged vCyclin. WCLs were prepared at 72 h and analyzed by 2DGE and immunoblotting. **F** 293 T cells were transfected with plasmids expressing Myc-vCyclin. WCLs were prepared at 48 h and analyzed by 2DGE and immunoblotting. **G** 293 T cells were transfected with plasmids expressing CAD with or without vCyclin for 48 h. CAD was then purified and analyzed by Coomassie staining (right panel). In vitro deamidation was performed with purified GST-RelA for 15 min, which was analyzed by 2DGE (left panel). **H** 293 T cells were transfected with the NF-κB reporter cocktail, a plasmid containing RelA, and increasing amounts of a plasmid containing vCyclin or CDK6. NF-κB activation was determined by luciferase assay at 30 h. **I** HOKs were infected with lentivirus containing vCyclin. Intracellular metabolites were extracted, and lactate was quantified by mass spectrometry. **J** 293 T cells were depleted of CAD by CAD-specific shRNA. The stable cells were then infected with lentivirus containing vCyclin. The culturing media were collected to determine lactate concentration at 16 h post medium replacement. **K** HOKs were infected with lentivirus containing vector control or vCyclin. Extracellular acidification rate (ECAR) was measured by Seahorse analyzer. Data are presented as mean ± SD of n = 3 biological replicates (2H-2K). Blots were representative of at least two independent experiments (2A-2C, 2E-2G). Significance was calculated using two-tailed, unpaired Student's t-test. Source data are provided as a source data file. See also Fig. S2.

Ser-1900 phosphorylation in 293 T cells, but it failed to recognize a phospho-mimetic mutant (S1900D) and phospho-null mutant (S1900A) of CAD (Fig. S4F). Using this antibody, we validated that S1900 phosphorylation increased following CAD incubation with CDK6 and ATP in vitro (Fig. 4F). Furthermore, WT KSHV infection potently enhanced CAD S1900 phosphorylation, whose effect was minimal during ΔCyclin infection (Fig. 4G). In line with this, depletion of CDK6 abolished the S1900 phosphorylation in both mock-treated and KSHV-infected cells (Fig. 4H). Finally, vCyclin overexpression was sufficient to induce CAD S1900 phosphorylation and the expression of RelA deamidation-dependent gene *HK4*, while such effects were ablated upon CDK6 knocking down (Figs. S4G and S4H). Altogether, these data support a model in which KSHV vCyclin recruits CDK6 to phosphorylate CAD at Ser-1900.

## S1900 phosphorylation activates CAD

To define the impact of S1900 phosphorylation on the enzymatic activities of CAD, we characterized the CAD phospho-mimetic mutant (S1900D) and phospho-null mutant (S1900A) functionally. We purified wild-type (WT) CAD and its mutants from transfected 293 T cells (Fig. 5A, right) and performed in vitro deamidation assay. This analysis demonstrated that S1900D potently induced RelA deamidation as compared to WT, while S1900A showed reduced activity (Fig. 5A). In parallel, in vitro enzymatic assay showed increased glutaminase activity for S1900D as compared to WT. In contrast, S1900A had a diminished activity (Figs. 5B and S5A). Notably, CAD S1859E, a mimetic mutant of the previously reported CAD phosphorylation (S1859) that regulates CAD activity[39,40], showed a comparable glutaminase activity to WT in vitro (Fig. 5B).

Besides in vitro biochemical activity alterations by S1900 phosphorylation, we also observed an increased [$^{15}$N]-glutamine incorporation into dihydroorotic acid and an upregulated steady state level for dihydroorotic acid in 293 T CAD knockout (KO) cells[34] reconstituted with S1900D as compared to cells expressing WT or S1900A (Fig. 5C and D). In parallel, deamidation of endogenous RelA was promoted by exogenous S1900D, while S1900A showed a reduced deamidation activity compared to WT (Fig. 5E). Corresponding to the deamidase activity enhancement by S1900 phosphorylation, S1900D ectopic expression demonstrated a stronger inhibition on Sendai virus or RelA-induced NF-κB responses (Fig. S5B and C), as well as a higher expression of deamidation-dependent metabolic genes compared to WT or S1900A (Fig. S5D). In cells depleting of endogenous CAD (Fig. S5E), reconstitution of S1900D elevated lactate production (Fig. 5F) and boosted cell proliferation (Fig. 5G) as compared to those expressing vector control, WT, or S1900A. Finally, when HOKs reconstituted with the CAD WT, S1900D or S1900A mutant (Fig. S5F) were infected with KSHV, the S1900D-reconstituted cells produced a comparable amount of infectious virions as cells expressing WT, suggesting that WT was potently phosphorylated by KSHV vCyclin. However, the viral titers of cells expressing CAD S1900A cells were significantly decreased (Fig. 5H). Altogether, our data demonstrated that vCyclin-CDK6 phosphorylates CAD at Ser-1900, thereby exploiting CAD for de novo pyrimidine synthesis and deamidation-mediated metabolic reprogramming, both of which contribute to cell proliferation and viral replication.

## CAD and RelA deamidation are required for KSHV pathogenesis

We next sought to dissect the role of CAD and RelA deamidation in regulating KSHV pathogenesis. Our previous findings demonstrated that NF-κB activation potently inhibits KSHV lytic replication and drives latency[47], implicating that deamidation of RelA may have dual functions for KSHV lytic replication: (1) fine-tuning NF-κB responses to promote KSHV gene transcription; and (2) driving glycolysis to provide building blocks for virion production. To test this hypothesis, we depleted endogenous RelA from HOKs with a RelA-specific shRNA targeting 3'UTR, and then reconstituted these cells with either empty vector, a fully-deamidated RelA mutant DD (N64D and N139D), or a deamidation-resistant RelA mutant 64A (N64A) (Fig. S6A)[34]. Compared to vector-expressing cells, the mRNA expressions of KSHV viral genes *LANA*, *PAN*, and *ORF57* increased in HOK-DD cells (Fig. 6A). In stark contrast, *PAN* and *ORF57* transcripts were decreased in the HOK-64A cells upon KSHV lytic replication (Fig. 6A). HOK-DD HOKs also produced more viral particles compared to control and 64A-expressing cells (Fig. 6B). Upon establishing latency, we observed a reduction of LANA mRNA expression in RelA-64A reconstituted TIME cells compared to the cells reconstituted with RelA-DD (Fig. 6C). Moreover, depletion of endogenous CAD by shRNAs and CRISPR-Cas9/sgRNAs in LECs (Fig. S6B) and HOKs (Fig. S6C and S6D) resulted in a marked decrease in viral lytic gene expression (Figs. S6E, F, and 6D) and viral titers (Fig. 6E–H). In accord, our data showed that the deamidation-resistant mutant 64 A reconstituted TIME cells were unable to induce deamidation-dependent metabolic gene expression (Fig. S6G) and to drive glycolysis (Fig. 6I) upon KSHV latent infection. A similar phenotype was observed upon depletion of CAD, which led to reduced expressions of deamidation-dependent genes (Fig. S6H) in LECs and a partial abrogation of the boosted effect on glycolysis (Fig. 6J) in HOKs during KSHV infections. Taken together, our data indicated that both CAD activity and RelA deamidation are necessary for KSHV-mediated glycolysis reprogramming and KSHV viral lytic replication.

We next addressed the role of CAD in the proliferation and tumorigenesis of KSHV latent-infected endothelial cells and lymphoma cells. A CAD ATCase specific inhibitor, N-phosphonacetyl-l-aspartate (PALA[48]), preferentially inhibited the growth of KSHV-infected TIME cells compared to the mock-infected control (Fig. 6K), emphasizing the importance of CAD for KSHV to drive cell survival and proliferation. In the KSHV-positive primary effusion lymphoma (PEL) cell line BCBL-1, depletion of CAD by CRISPR/Cas9 near-abolished cell proliferation (Figs. S6I and 6L) in vitro, which can be partially rescued by the expression of RelA-DD (Fig. S6J). When a single clone of *CAD*$^{-/-}$

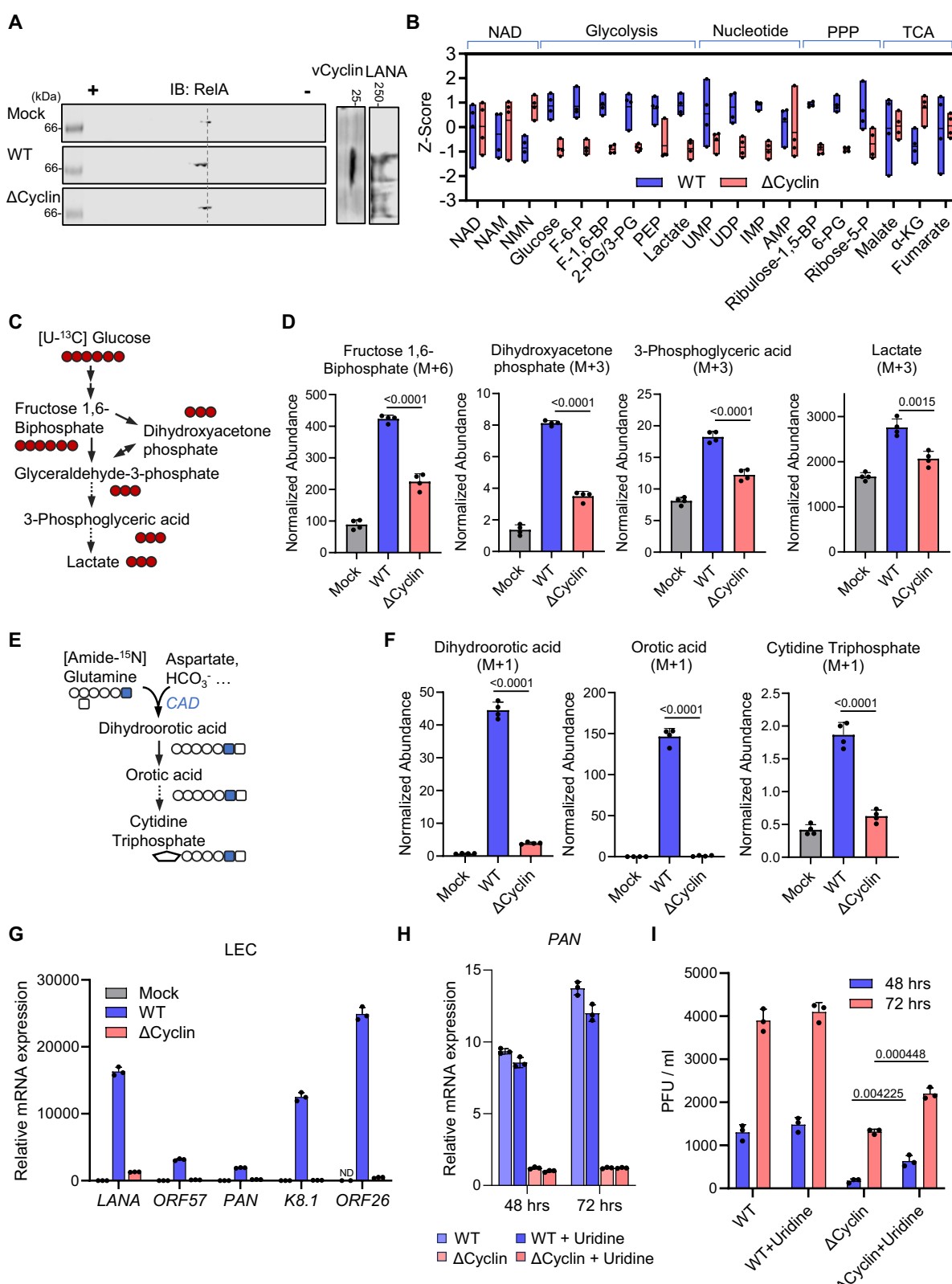

BCBL-1 was amplified, it failed to propagate in vivo upon engraftment into the NOD-SCID mice (Fig. 6M). Similarly, CAD was required for the optimal growth of another PEL cell line BC-3 (Fig. S6K and L). Finally, we observed a partial decrease of cell proliferation upon CAD knockout in KSHV and Epstein-Barr virus (EBV) double positive PEL cell line JSC-1 (Figs. S6M and 6N). Nevertheless, the single clone of *CAD*[-/-] JSC-1 failed to form solid tumors in the engrafted NOD-SCID mice (Fig. 6O),

demonstrating the pivotal role for CAD in driving γ−herpesvirus-associated malignancy in vivo.

## CDK6 and CAD inhibitors inhibit KSHV lytic replication and tumorigenesis

Based on the observation that KSHV exploits CDK6 to activate CAD for metabolic reprogramming and tumorigenesis, we investigated the

**Fig. 3 | Metabolic reprogramming by KSHV vCyclin. A** HOKs were infected with KSHV-wild-type (WT) or KSHV-vCyclin.STOP (ΔCyclin) (MOI = 30) for 48 h. WCLs were prepared and analyzed by 2DGE and immunoblotting. **B** HOKs were infected with KSHV-WT or KSHV-ΔCyclin (MOI = 30) for 48 h. Intracellular metabolites were extracted. NAD, glycolysis, nucleotide, pentose phosphate pathway (PPP), and TCA cycle intermediates were quantified by mass spectrometry and normalized by z-score. Data were presented as floating bars of n = 4 biological replicates. Bounds of box (bottom and top) represented min and max, and line represented mean. Full names of the metabolites are provided in the methods. **C**, **D** A diagram of the isotope labeling of glycolytic intermediates using [U-$^{13}$C] glucose (**C**). Mass spectrometry analysis of intracellular glycolysis intermediates was performed at 15 min after labeling the HOKs pre-infected with KSHV-WT or KSHV-ΔCyclin (D). (M + a) indicates the mass of universally labeled glycolysis intermediates ( + a) with [U-$^{13}$C] glucose. **E**, **F** A diagram of the isotope labeling of de novo pyrimidine synthesis intermediates using [Amide-$^{15}$N] glutamine (**E**). Mass spectrometry analysis of pyrimidine intermediates was performed at 15 min after labeling the HOKs pre-infected with KSHV-WT or KSHV-ΔCyclin (**F**). (M + 1) indicates the mass of labeled pyrimidine intermediates ( + 1) with [Amide-$^{15}$N] glutamine. **G** RT-qPCR analysis of the indicated viral mRNAs in LECs infected with KSHV-WT or KSHV-ΔCyclin (MOI = 10) for 72 h. ND: undetected. **H** RT-qPCR analysis of *PAN* mRNA in HOKs infected with KSHV-WT or KSHV-ΔCyclin (MOI = 30) for the indicated hours with or without uridine (10 μg/ml). **I** Viral titer in the medium was determined for the infected HOKs as shown in (**H**). Data are presented as mean ± SD of n = 4 biological replicates (3D and 3 F) and n = 3 biological replicates (3G-3I). Blots were representative of at least two independent experiments (3 A). Significance was calculated using two-tailed, unpaired Student's t-test. Source data are provided as a source data file. See also Fig. S3.

therapeutic potential of targeting CDK6 and CAD in blocking KSHV viral pathogenesis. Palbociclib[49], an FDA-approved CDK4/6 inhibitor for breast cancer, blocked S1900 phosphorylation of CAD in KSHV de novo infected HOKs (Fig. 7A) and KSHV-positive BCBL-1 lymphoma cells (Fig. 7B). When examining endogenous RelA by 2DGE we found that Palbociclib treatment abolished RelA deamidation induced by KSHV (Fig. 7C). Furthermore, Palbociclib and 6-Diazo-5-oxo-L-norleucine (DON)[50], a CAD glutaminase (deamidase) inhibitor, blocked the upregulation of deamidation-dependent metabolic gene expression (Fig. 7D) and lactate production by KSHV (Figs. 7E and S7A). Chemical inhibition of CAD and CDK6 activity also inhibited KSHV lytic gene expression (Fig. 7F) and virion production in both LECs and HOKs (Figs. 7G, S7B and C). These results collectively showed that CDK6 and CAD inhibitors impede KSHV-mediated RelA deamidation, glycolysis reprogramming, and lytic replication.

To test whether targeting CAD and CDK6 thwarts the progression of KSHV-associated primary effusion lymphomas and potentially virus-negative lymphomas, we treated four lymphoma cell lines, BJAB (KSHV-negative Burkitt-like Lymphoma), as well as PEL cells BC-3 (KSHV-positive), BCBL-1 (KSHV-positive), and JSC-1 (KSHV- and EBV-positive), with PALA, DON, and Palbociclib. PALA blocked cell proliferation of these cells with IC50s of 48 ~ 216 μM (Fig. S7D), while DON treatment resulted in a similar inhibitory effect against all four cell lines (Fig. 7H) with IC50 < 2 μM. Virus-positive cells also demonstrated increased sensitivities to Palbociclib compared to virus-negative BJAB cells (Fig. 7I), which was consistent with previous reports[51,52] and suggested that CDK4/CDK6 may have additional functions to support the survival and proliferation of virus-transformed lymphoma cells. BC-3 cells expressing RelA-DD showed resistance to Palbociclib, whereas addition of uridine further conferred drug resistance (Fig. S7E), emphasizing the metabolic functions of CDK6 in BC-3 to support cell proliferation. With mock-infected and KSHV-infected BJAB cells, we found that DON and PALA preferentially inhibited cell proliferation for BJAB-KSHV at lower doses (Fig. 7J). Lastly, treatment of DON and PALA (at 400 μM) markedly blocked the colony formation of BC-3 and BCBL-1 cells, while BJAB and JSC-1 demonstrated resistance (Figs. S7F–I and S8A, B). On the other hand, Palbociclib effectively inhibited the colony formation for all four cell lines (Fig. S7F–I).

Next, we evaluated the efficacy of Palbociclib and JHU083, a prodrug of DON designed to be activated within the tumor microenvironment[23], to inhibit tumorigenesis in vivo. Specifically, BC-3-luciferase, BCBL-1-luciferase, and JSC-1-luciferase cells were intraperitoneally transplanted into the NOD-SCID mice and drugs were administered upon steady establishment of the tumor mass determined by IVIS imaging. Remarkably, JHU083 or Palbociclib treatment resulted in an undetectable tumor burden for BCBL-1 tumors (Figs. 7K and S8C). The treatments of JHU083 correlated with a lack of weight gain, which is typically caused by abdominal ascites accumulation (Fig. S8D), and a 100% survival (Fig. S8E). In comparison, JHU083 significantly thwarted tumor progression of BC-3, evidenced by a marked reduction of luminescence (Fig. S8F) and improved survival (Fig. 7L), whereas Palbociclib showed a partial tumor-inhibiting effect. Different than BC-3 and JHU083, JSC-1 cells form solid tumors upon engraftment. Treatment of both Palbociclib and JHU083 significantly reduced luminescence signals (Fig. S8G), and led to a ~ 80% reduction in tumor size and weight (Fig. 7M and N). Collectively, our data indicate that targeting CDK6 and CAD is a promising strategy against γ-herpesvirus-induced primary effusion lymphomas.

## Discussion

Viruses rely on the host metabolic machinery for survival and replication. Despite carrying a few genes with metabolic functions, KSHV largely lacks its own metabolic apparatus to produce essential building blocks, such as nucleotides, carbohydrates, and lipids. As such, KSHV has evolved to manipulate critical host metabolic pathways, including aerobic glycolysis, glutaminolysis, polyamine biosynthesis, and lipogenesis[12–20,53,54]. However, the mechanisms underlying viral manipulations of cellular metabolism remain poorly understood. Interestingly, it has been reported that glutamine intake increases in KSHV-transformed cells, which provides γ-nitrogen for purine and pyrimidine synthesis that are required for cell survival and proliferation[17]. In this study, we identified that KSHV facilitated de novo pyrimidine synthesis by exploiting the γ-nitrogen-utilizing enzyme CAD. Our work suggests that activated CAD may couple with the upregulated glutamine consumption to create an accelerated anabolic state for viral replication and cell proliferation. Furthermore, our previous research demonstrated that CAD serves as an active player in governing aerobic glycolysis via its non-canonical deamidase activity on the NF-κB transcription factor RelA[34]. Correspondingly, KSHV manipulation of CAD contributes to glycolysis upregulation. Notably, the level of the glycolysis upregulation varied among lactate assays and Seahorse assays, likely due to distinct cell types, experimental settings, and media composition. Depletion of CAD impaired (but not abolished) the glycolysis upregulation induced by KSHV, suggesting that other mechanisms exist for the virus to drive glycolysis, as were reported by previous studies[14,15,55].

Metabolic reprogramming via regulatory PTMs on rate-limiting enzymes represents one of the hallmarks of cell proliferation and cancer[22]. CAD has previously been shown to undergo phosphorylation at S1859 by S6K following mTORC1 activation[39,40]. Of note, this mechanism is hijacked by SARS-COV-2 NSP9 to promote nucleotide biosynthesis[56]. Ben-Sahra et al. also reported the detection of CAD peptides containing S1900 phosphorylation[40]; however, unlike S1859, S1900 phosphorylation remained unchanged upon insulin or rapamycin treatment[40], suggesting that this site is not dependent on nutrient sensing. In this study, we identified CDK6 as the kinase mediating S1900 phosphorylation, which is driven by a KSHV homologue of mammalian D-type cyclin. Based on our study, we hypothesize that while mTORC1-S6K drives pyrimidine synthesis in response to nutrient signals, host may deploy CDK6 to activate CAD during cell

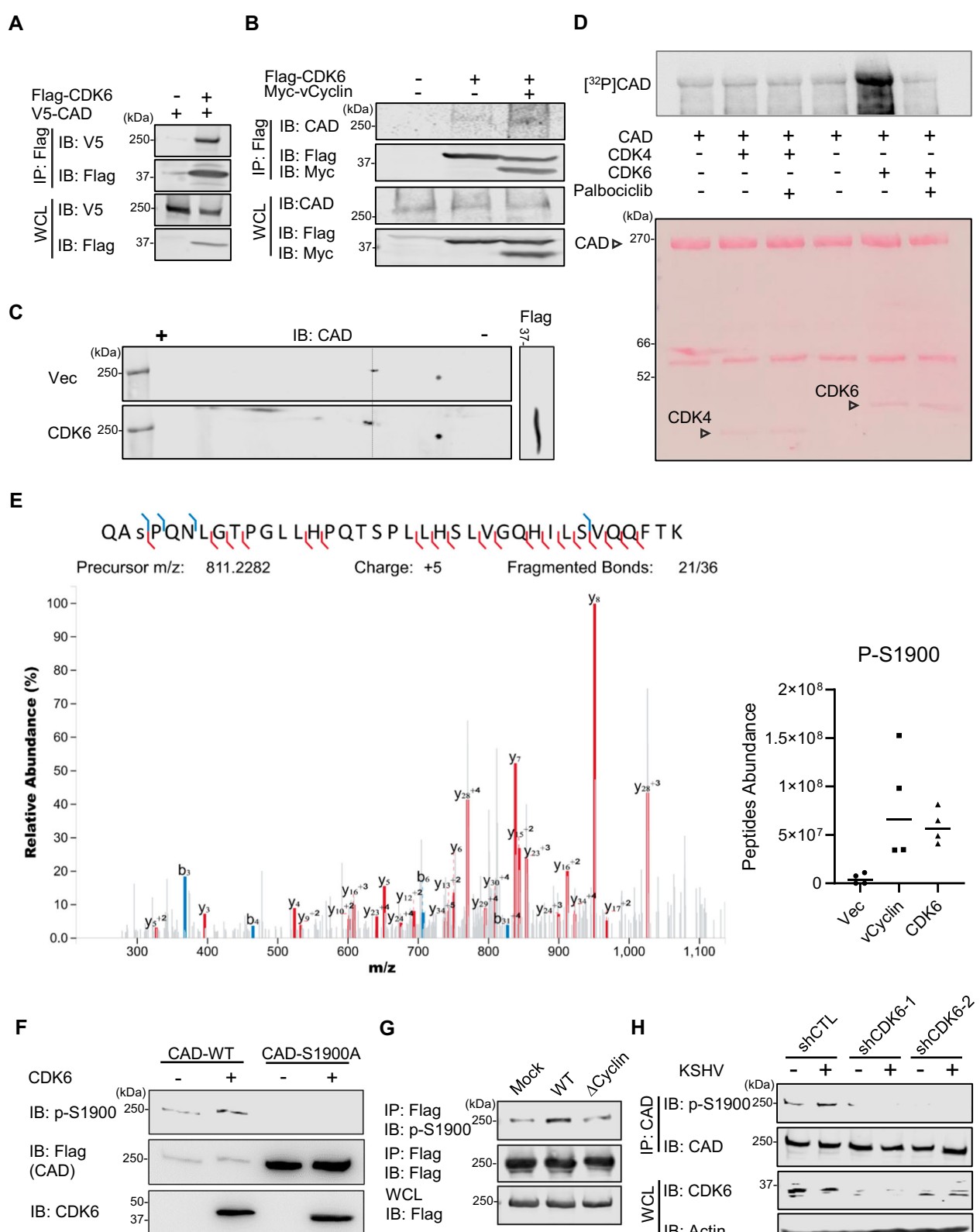

cycle progression to meet the metabolic demand of DNA replication and cell division. Thus, CDK6-mediated CAD activation and RelA deamidation may serve as a host intrinsic metabolic reprogramming mechanism that is exploited by the virus. In support of this, inhibiting CAD or CDK6 reduced glycolysis in LECs in the absence of KSHV infection. Moreover, RelA deamidation and CAD deamidase activity are dynamically regulated during cell cycle progression[34]. While KSHV

deploys vCyclin which hijacks CDK6 to constitutively activate CAD regardless of cell cycle, it is reasonable to propose that non-viral cancer cells (such as BJAB) may have evolved similar mechanisms to exploit CAD for anabolic metabolism and tumorigenesis via dysregulated Cyclin Ds, potentially making CAD an Achilles' heel for them. While our preliminary results support CAD as a metabolic vulnerability for KSHV-infected cells, extensive future research will be needed to

**Fig. 4 | CDK6 phosphorylates CAD at S1900. A** 293 T cells were transfected with plasmids expressing Flag-CDK6 and V5-CAD for 48 h. WCLs were precipitated with anti-Flag (CDK6). Precipitated proteins and WCLs were analyzed by immunoblotting with the indicated antibodies. **B** 293 T cells were transfected with plasmids expressing Flag-CDK6 with or without vCyclin for 48 h. WCLs were precipitated with anti-Flag (CDK6). Precipitated proteins and WCLs were analyzed by immunoblotting with the indicated antibodies. **C** 293 T cells were transfected with a plasmid expressing Flag-CDK6 for 48 h. WCLs were prepared and analyzed by 2DGE and immunoblotting. **D** In vitro [γ-³²P]-ATP kinase assay was performed on CAD with purified CDK4 or CDK6 and visualized by autoradiography (Top). The membrane was then stained with Ponceau S (Bottom). Palbociclib: a CDK4/CDK6 inhibitor. **E** The m/z spectra of the peptide containing S1900 phosphorylation was shown (Left), with its abundance in 293 T cells ectopically expressing an empty vector, vCyclin, or CDK6 (Right). **F** In vitro ATP kinase assay was performed on CAD wild-type (WT) and S1900A mutant with purified CDK6, which was then analyzed by SDS-PAGE and immunoblotting with the indicated antibodies. **G** 293 T cells were transfected with a plasmid expressing Flag-CAD and then infected with KSHV-wild-type (WT) or KSHV-vCyclin.STOP (ΔCyclin) (MOI = 5) for 24 h. WCLs were precipitated with anti-Flag (CAD). Precipitated proteins and WCLs were processed in parallel and analyzed by immunoblotting with the indicated antibodies. **H** 293 T cells were depleted of CDK6 by CDK6-speific shRNA and then infected with KSHV (MOI = 5) for 24 h. WCLs were precipitated with anti-CAD. Precipitated proteins and WCLs were processed in parallel and analyzed by immunoblotting with the indicated antibodies. Data are presented as mean ± SD of n = 4 biological replicates (4E). Blots were representative of at least two independent experiments (4A-4D, 4F-4H). Source data are provided as a source data file. See also Fig. S4.

assess the association between a dysregulated CAD and cell susceptibility to CAD inhibitors in more types of cancers.

We observed a marked increase of CAD glutaminase activity for S1900D compared to WT, whereas S1900A demonstrated significantly lower catalytic ability, suggesting that S1900 phosphorylation promotes CAD activity but does not serve as an on-off switch for CAD enzymatic functions. Interestingly, we observed comparable in vitro enzymatic activity between WT and S1859E. Both S1900 and S1859 locate at the linker region that joins Dihydroorotase (DHO) domain and Aspartate Transcarbamoylase (ATC) domain of CAD. A previous study demonstrated that S1859 phosphorylation promotes CAD oligomerization[39], a hallmark of CAD activation. We think that the effect of S1859E on CAD oligomerization may have been neutralized in our in vitro experimental settings with purified proteins. Meanwhile, the difference on CAD glutaminase activity between S1900 and S1859 phosphorylation suggests their distinct mechanisms to drive CAD activation. A previous study showed that deletion of the linker region decreases 'substrate channeling' of the enzyme[57]. As a feature of multifunctional enzymes which regulates kinetics, substrate channeling refers to the direct delivery of the reaction intermediate within the enzyme without dissociation into the environment. One possibility is that S1900 phosphorylation may enhance the 'substrate channeling' of carbamoyl phosphate, thereby accelerating all the functional domains including the glutaminase. The detailed molecular mechanism underlying CAD activation by S1900 phosphorylation requires further investigation. Additionally, extensive experimentation and mass spectrometry analyses will be needed to probe whether additional phosphorylation site(s) are present in CAD that are mediated by CDK6, which may synergize with S1900 phosphorylation to regulate CAD activities. Finally, in our isotope tracing experiments we incubated the cells with tracer for a short period of time, thereby enriching ¹⁵N-labeled CAD reaction product (dihydroorotic acid) to define the direct effect of KSHV on the activity of CAD. This also limited the window of analysis and potentially diminished the differences among mock, WT, or ΔCyclin-infected cells for downstream pyrimidine products (e.g., CTP). Future experiments with different time points will be needed to delineate the reprogramming of whole de novo pyrimidine synthesis pathway by vCyclin-CDK6-mediated CAD phosphorylation.

Recent studies have reported cell cycle-independent functions of CDK4 and CDK6[58,59]. Importantly, both CDK4 and CDK6 are involved in the regulation of central carbon metabolism[60,61]. The cyclin D3-CDK6 complex has been shown to reprogram glycolysis, which ultimately favors the pentose phosphate pathways (PPP) and serine biosynthesis, thereby removing reactive oxygen species in human T cell acute lymphoblastic leukemia (T-ALL) cells[58]. Opposite to its role in leukemia cells, CDK6 was found to drive glycolysis in cervical cancer cells[60], which is in line with our findings that CDK6 promotes CAD-mediated RelA deamidation to drive glycolysis in virus-infected oral keratinocytes and endothelial cells. These observations indicate that CDK6-mediated glycolysis reprogramming is likely cell type- and tissue-

specific. In this study, we further identified a metabolic role of CDK6 to activate pyrimidine synthesis via CAD phosphorylation. Although purified CDK4 was not sufficient to phosphorylate CAD in vitro, within the host-cell mammalian Cyclin Ds may activate both CDK4 and CDK6 to drive CAD phosphorylation[62] at S1900 for nucleotide synthesis and glycolysis reprogramming. Therefore, our study implicates that energy metabolism and macromolecule biosynthesis may be tightly coordinated by CDK6 and its related CDKs to drive cell proliferation. Importantly, reconstitution of RelA-DD and uridine supplementation failed to fully restore the replication of KSHV-ΔCyclin, and we observed a broader metabolic reprogramming in WT KSHV, but not ΔCyclin-infected cells. These results indicate that vCyclin and CDK6 may have additional mechanisms to control metabolism and promote viral lytic replication. On the other hand, as CDK6 regulates cell cycle progression and potentially other physiological functions, depletion of CDK6 or inhibition of its activity may have direct and indirect effects on cellular metabolism, which complicates the phenotype observed in KSHV-infected or vCyclin-transduced cells. Further investigation to disrupt vCyclin-CAD interaction, as well as to probe KSHV-driven metabolic reprogramming in cells reconstituted with WT or S1900A mutant of CAD, will help delineate vCyclin-CDK6-CAD signaling without disrupting normal cellular physiology. In summary, a deepened understanding of the metabolic roles of CDK4/6 will open up avenues for the diagnosis of and metabolic interventions against KSHV malignancies and cancers with dysregulated Cyclin D/CDK4/CDK6 activities.

In primary effusion lymphoma cells, vCyclin is reported to be stably expressed throughout the cell cycle, resulting in a constitutively activated vCyclin/CDK6 complex[63]. On the other hand, blocking S phase entry by Palbociclib reduced viral replication upon reactivation[64]. Moreover, it has been shown that the proliferation of specific PEL cells is also dependent on cellular Cyclin D2/CDK4 and Cyclin D2/CDK6[51]. We also observed potent S1900 phosphorylation in KSHV-infected and transformed cells, which is abrogated by CDK6 depletion or chemical inhibition. Based on our findings, blocking CDK4/CDK6 not only prevents Rb1 phosphorylation leading to cell cycle arrest, but also inhibits the metabolic functions of vCyclin and likely cellular Cyclin Ds. As such, repurposing of CDK4/CDK6 inhibitors (such as Palbociclib) to treat KSHV-associated cancers may target multiple signaling pathways which will minimize drug resistances. Our in vivo xenograft data also showed the potent efficacy of targeting CAD to block the tumorigenesis of multiple PEL cells. Importantly, JHU083, as a DON prodrug, blocks the glutaminase (deamidase) activity of CAD and overcomes the toxicity issues of DON by selectively targeting tumor microenvironment[23], demonstrating its therapeutic potential for future clinical evaluation. Notably, DON also targets other glutamine amidotransferases[50], making its effect not solely due to CAD inhibition. On the other hand, as a CAD specific inhibitor, PALA[65] is not selective against tumors[66] and targets the ATC domain of CAD which may not block CAD-mediated RelA deamidation. As such, tumor-targeting CAD specific inhibitors that inhibit both protein deamidation

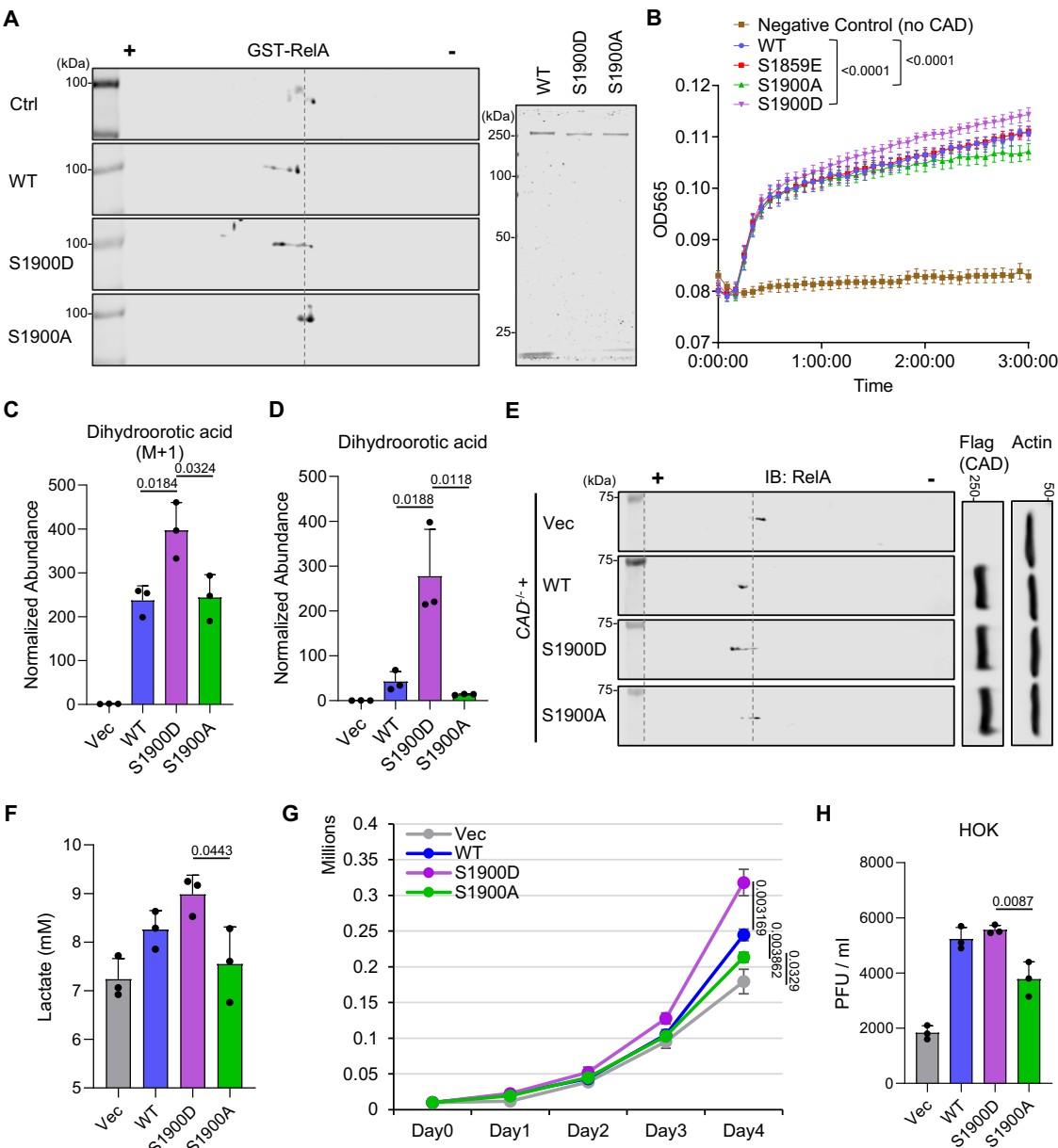

**Fig. 5 | CAD S1900 phosphorylation promotes metabolic reprogramming. A** In vitro deamidation was performed on purified GST-RelA with purified CAD wild-type (WT), CAD S1900D mutant (S1900D), or CAD S1900A mutant (S1900A) (Right), and analyzed by 2DGE with anti-GST antibody (Left). **B** CAD WT, S1900D, S1900A, and S1859E were purified from ectopically expressed 293 T cells. In vitro enzymatic (glutaminase) assay was performed. Glutamate concentration was quantified at 565 nm. **C** *CAD⁻/⁻* 293 T cells were reconstituted with CAD WT, S1900D, or S1900A by transient transfection for 48 h. Mass spectrometry analysis of ¹⁵N-labeled-dihydroorotic acid was performed at 15 min after labeling the cells with [Amide-¹⁵N] glutamine. **D** Mass spectrometry analysis of intracellular dihydroorotic acid was performed on the reconstituted *CAD⁻/⁻* 293 T cells as shown in (**C**). **E** WCLs were prepared from the reconstituted *CAD⁻/⁻* 293 T cells as shown in (**C**) and analyzed by 2DGE and immunoblotting. **F** 293 T cells were depleted of CAD by CAD-specific

shRNA targeting 3′UTR. The stable cells were then transfected with plasmids expressing CAD WT, S1900D, or S1900A. The culturing media were collected from 1 ×10⁶ cells to determine lactate concentration at 20 h post medium replacement. **G** 1 ×10⁴ of the reconstituted 293 T cells as shown in (**F**) were seeded and cultured for a continuous 4 days and cell numbers were counted. **H** HOKs were depleted of CAD by CAD-specific shRNA targeting 3′UTR. The stable cells were then transfected with plasmids expressing CAD WT, S1900D, or S1900A, before infection with KSHV (MOI = 30). The viral titers in the culturing medium were determined 72 h post infection. Data are presented as mean ± SD of n = 3 biological replicates (5 C, 5D, 5F-5H) and mean ± SEM of n = 8 biological replicates (5B). Blots were representative of at least two independent experiments (5 A and 5E). Significance was calculated using two-tailed, unpaired Student's t-test (paired t-test for 5B). Source data are provided as a source data file. See also Fig. S5.

and de novo pyrimidine synthesis remain to be developed for metabolic interventions against KSHV-associated malignancies.

In summary, our study uncovers a mechanism underlying the exploitation of host nucleotide biosynthesis and glycolysis by an oncogenic herpesvirus to support viral lytic replication and tumorigenesis, and unveiled two druggable molecular targets (CDK6 and CAD) against KSHV-associated malignancies in vivo.

# Methods

## Ethics statement

All cell and virus experiments were performed in Biosafety Level 2 or 2 plus facilities with the approval by Institutional Biosafety Committee (IBC) at Cleveland Clinic FRIC. All animal work was performed under strict accordance with the recommendation in the Guide for the Care and Use of Laboratory Animals of the National Institutes of Health. The

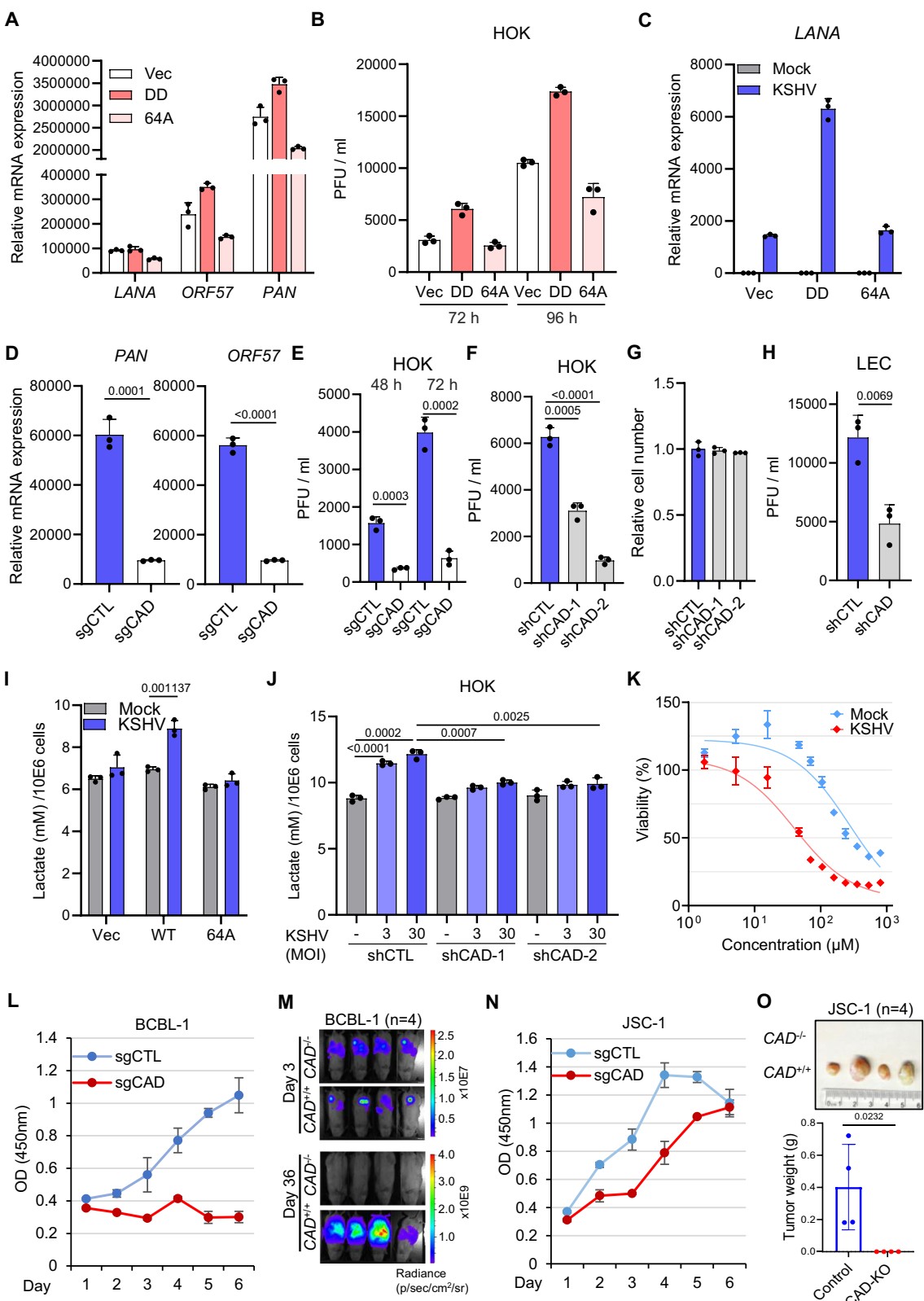

animal protocol was approved by the Institutional Animal Care and Use Committee (IACUC) of Cleveland Clinic FRIC.

## Cell culture

HEK293T was purchased from ATCC. HOK16B, TIME, BCBL-1, BC-3, JSC-1, and BJAB cells were generously provided by Dr. Pinghui Feng (University of Southern California) and Dr. Ren Sun (University of California Los Angeles). ISLK.BAC16-WT and iSLK.BAC16-deltaCyclin were generously provided by Dr. Jae U Jung (Cleveland Clinic) and Dr. Michael Lagunoff (University of Washington). LECs were generously provided by Dr. Young-kwon Hong (University of Southern California). Cells were maintained at 37 °C in a humidified incubator with 5% $CO_2$. HEK293T and iSLK-BAC16 cells were cultured in Dulbecco's modified Eagle's medium (DMEM, Hyclone) supplemented with 10%

**Fig. 6 | CAD and RelA deamidation for KSHV pathogenesis. A** HOKs depleted of RelA were reconstituted with RelA-DD or RelA-64A by lentiviral transduction, then infected with KSHV (MOI = 30) for 72 h. RT-qPCR analysis of viral mRNAs was performed. **B** The viral titers in the media of the cells in (**A**) at the indicated hours. **C** TIME depleted of RelA were reconstituted with RelA-DD or RelA-64A by lentiviral transduction, then infected with KSHV (MOI = 3) for 48 h. RT-qPCR analysis of *LANA* mRNAs was performed. **D, E** RT-qPCR analysis of viral mRNAs in HOKs depleted of CAD by CRISPR-Cas9/sgRNA and infected with KSHV (MOI = 30) for 72 h (**D**). Viral titers in the media were determined (**E**). **F, G** Viral titers in the media of HOKs depleted of CAD by CAD-specific shRNA and infected with KSHV (MOI = 30) for 72 h (**F**). Cell numbers were counted and normalized (**G**). **H** Viral titers in the media of LECs depleted of CAD and infected with KSHV (MOI = 10) for 72 h. **I** TIME cells depleted of RelA were reconstituted with RelA-WT or RelA-64A, then infected with KSHV (MOI = 3) for 48 h. Lactate in the media was determined at 20 h post medium replacement. **J** Lactate in the media at 24 h post medium replacement in HOKs depleted of CAD and infected with KSHV for 48 h. **K** Viability of mock and KSHV-infected TIME cells in the presence of N-phosphonacetyl-l-aspartate (PALA). **L** Proliferation of BCBL-1 depleted of CAD by CRISPR-Cas9/sgCAD. **M** Single clone of *CAD^+/+^* and *CAD^-/-^* BCBL-1-luciferase was amplified. 5 ×10^6 of cells were engrafted into NOD-SCID mice and IVIS imaging was performed. **N** Proliferation of JSC-1 depleted of CAD by CRISPR-Cas9/sgCAD. **O** Single clone of *CAD^+/+^* and *CAD^-/-^* JSC-1-luciferase was amplified. 1 ×10^7 of cells were engrafted into NOD-SCID mice and solid tumors were extracted and weighed at day 35. Data are presented as mean ± SD of n = 3 biological replicates (6A-6L, and 6 N). Significance was calculated using two-tailed, unpaired Student's t-test. Source data are provided as a source data file. See also Fig. S6.

fetal bovine serum (Gibco) and 1% penicillin/streptomycin (Gibco). ISLK-BAC16 cells were cultured in the presence of 1 μg/mL of puromycin, 250 μg/mL of G418, and 1200 μg/mL of hygromycin. TIME and LEC cells were cultured in VascuLife VEGF endothelial medium (Lifeline). HOK16B cells were cultured in KGM-2 keratinocyte growth medium (Lonza). The primary effusion lymphoma (PEL) cell lines (BCBL-1, BC-3, and JSC-1) and KSHV-negative lymphoma cell line BJAB were cultured in RPMI 1640 medium supplemented with 10% FBS and 1% penicillin/streptomycin. ISLK.BAC16 cell lines (derived from a renal-cell carcinoma cell line Caki-1[67]) were only used for KSHV production in this study.

Cells from ATCC were authenticated by the vendor. Cell lines which were obtained from and validated by other investigators were not further authenticated. ISLK.BAC16-WT and iSLK.BAC.16-deltaCyclin cell lines were validated by KSHV production and immunoblotting for vCyclin expression upon KSHV infection. The presence of KSHV in BCBL-1, BC-3, and JSC-1, as well as the absence of KSHV in BJAB were confirmed by RT-qPCR or immunoblotting. All cell lines for this study were tested negative for potential mycoplasma contamination by PCR-based Mycoplasma detection assays.

### Antibodies and reagents
Antibodies against GST (1:1000; Z-5, B-14, Santa Cruz), FLAG (1:1000; M2, Sigma), V5 (1:1000; D3H8Q, Cell Signaling), CAD (1:1000; A301-374A, Bethyl Laboratories), RelA (1:1000; 51-0500, Thermofisher), Pyruvate Carboxylase (1:1000; #66470, Cell Signaling, #49381, Cell Signaling), PDK3 (1:1000; AP7040a, Abcepta), Hexokinase 4 (1:1000; AP7901c, Abcepta), β-Actin (1:2000; 8H10D10, Cell Signaling), Phospho-CAD (1:1000; Ser1859, Cell Signaling), LANA (1:500; LN53, Abcam), vCyclin (1:500; 94B, Abcam), MYC(1:1000; 71D10, 9B11, Cell Signaling), CDK6 (1:1000; D4S8S, Cell Signaling) were purchased from the indicated suppliers. Polyclonal antibodies against KSHV vCyclin (1:500) and Phospho-CAD (Ser1900, 1:500) were generated by Genscript.

The glutamine analog 6-Diazo-5-oxo-L-norleucine (DON) was purchased from Sigma. JHU-083 (Ethyl 2-(2-Amino-4-methylpentanamido)-DON), Palbociclib, and N-(phosphonacetyl)-L-aspartic acid (PALA) were purchased from MedChemExpress. [U^13^C] Glucose and [Amide-^15^N] L-Glutamine were purchased from Cambridge Isotope Lab. Luciferin was purchased from Goldbio.

L-Lactate Assay Kit was purchased from Cayman Chemical. Cell proliferation Kit II, and Glutamate Assay Kit were purchased from Sigma. SYBR Green Master Mix was purchased from Bio-Rad and Genesee. Dual-Luciferase Reporter Assay System was purchased from Promega.

### Mice and in vivo study
Age-matched 6 to 14 weeks old female NOD.Cg-Prkdcscid/J (NOD-SCID, The Jackson Laboratory) mice were used for all tumor xenograft experiments. All animal work was performed under strict accordance with the recommendation in the Guide for the Care and Use of Laboratory Animals of the National Institutes of Health. The protocol was approved by the Institutional Animal Care and Use Committee (IACUC) of Cleveland Clinic FRIC. All mice were housed in a standard pathogen-free animal facility. Light cycle is 12:12 (on 7am, off 7 pm). Temperature range: 20-22 °C (68-72 °F). Humidity: 30-70%.

BCBL-1-luciferase, BC-3-luciferase, or JSC-1-luciferase cells (2.5-5 × 10^6, 1 × 10^7, or 1 × 10^7 cells/mouse, respectively) were intraperitoneally (i.p.) injected. For inhibitor studies, mice were randomized into vehicle and therapeutic groups. Three days post cells engraftment, the therapeutic groups were intraperitoneally injected once every two days with JHU-083 (Ethyl 2-(2-Amino-4-methylpentanamido)-DON) (7.4 mg/kg) or Palbociclib (9.5 mg/kg), while the vehicle control group mice were intraperitoneally injected with Lactated Ringer's solution. IVIS imaging was performed 1-2 times a week to measure the growth of tumor cells. For IVIS imaging, mice were anesthetized with 1.5% isoflurane in 100% oxygen. PBS-dissolved D-luciferin firefly potassium salt (Goldbio) was then injected intraperitoneally (150 mg/kg) before mice were imaged in the IVIS system (PerkinElmer) while in anesthesia. Data were presented as average radiance. General behavior of the mice was recorded, and body weight was measured daily to ensure good animal health was maintained. The experimental endpoint was set for mice with greater than 15% increase in body weight over a 7-day interval (20% increase in body weight over a 7-day interval as humane endpoint) with apparent ascites formation and abdominal distension for BC-3 and BCBL-1.

### KSHV propagation and infection
KSHV virus was prepared from iSLK-BAC16 cells as previously described[44]. Briefly, 70% confluent iSLK-BAC16 cells were induced with growth medium containing 1 mM sodium butyrate (Sigma) and 1 μg/mL doxycycline (Sigma). Four days later, culture medium containing cells and viral particles was harvested. Cells were then ruptured by repeated freeze-thaw cycles and then removed by centrifugation. Supernatant containing KSHV was concentrated by ultracentrifugation for 2 h at 100,000 g. Virus pellets were resuspended in DMEM medium and stored at −80 °C.

To determine the infectious virus titer, 0.1-0.5 ×10^6 HOK16B cells or 293 T cells were plated in a 48-well plate or 12-well plate and infected by KSHV with serial dilutions. The number of GFP positive cells was counted 48 h post infection and the virus titer was calculated.

For de novo infection, TIME cells, HOKs, or LECs of 70% confluence were infected with KSHV at a specific multiplicity of infection (MOI). The cell lines, time points, and MOIs were chosen and optimized based on published literatures[9,68,69]. The infected cells were spun at 750 g for 1 h at 30 °C and further incubated for 3 h at 37 °C before growth medium replacement. KSHV-infected TIME cells were selected with 500 μg/mL of hygromycin 48 h post infection to maintain KSHV latency.

### Plasmids and DNA transfection
Luciferase reporter plasmids for the NF-κB-Firefly luciferase, TK-Renilla luciferase, and RelA were described previously[26,27,70,71]. The non-

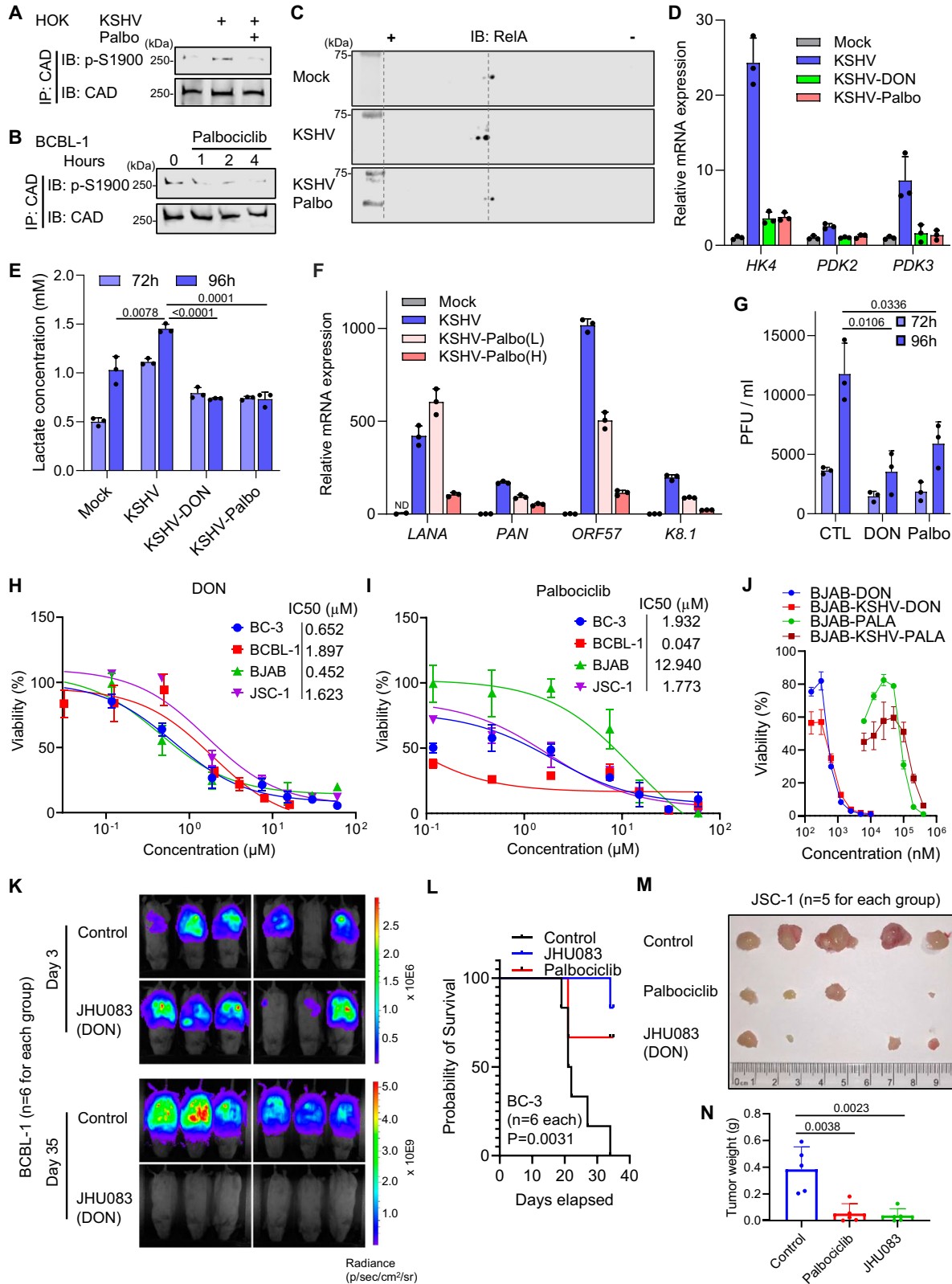

silencing (control) shRNA and shRNA for CAD and CDK6 were purchased from Thermo Scientific (GIPZ) and Sigma (Mission). Mammalian expression plasmids (pcDNA5) for CAD, vCyclin, and CDK6 were generated by molecular cloning based on their cDNA constructs. All gene mutants, including CAD truncation mutants, CAD-S1900D, CAD-S1900A, CAD-S1859E, RelA-N64A, and RelA-N64D/N139D (DD) were generated by site-directed mutagenesis and confirmed by sequencing. Lentiviral expression vectors for RelA mutants and vCyclin were generated in the vectors of pCDH-CMV-EF1-PURO or pCDH-CMV-EF1-Hygro by molecular cloning. SgRNAs for CAD were designed as previously described[34] and cloned into pLentiGUIDE-Puro or pLenti-CRISPR v2 (Addgene[72]).

**Fig. 7 | CDK6 and CAD inhibitors block KSHV pathogenesis. A** HOKs infected with KSHV (MOI = 30) for 48 h were treated with vehicle or Palbociclib for 4 h. WCLs were precipitated with anti-CAD. Precipitated proteins were analyzed by immunoblotting with the indicated antibodies. **B** BCBL-1 cells were treated with Palbociclib (50 nM). WCLs were precipitated with anti-CAD. Precipitated proteins were analyzed by immunoblotting with the indicated antibodies. **C** HOKs infected with KSHV (MOI = 30) for 48 h were treated with vehicle or Palbociclib for 4 h. WCLs were analyzed by 2DGE. **D** LECs infected with KSHV (MOI = 10) for 24 h were treated with DON (30 μM) or Palbociclib (2 μM) for 48 h. 72 h post infection, RT-qPCR analysis of the indicated mRNAs was performed. **E** LECs infected with KSHV (MOI = 10) for 24 h were treated with DON (30 μM) or Palbociclib (2 μM). Lactate in the media was determined at the indicated hours. **F** RT-qPCR analysis of viral mRNAs in LECs infected with KSHV (MOI = 10) for 24 h and then treated with Palbociclib (L: 0.4 μM, H: 2 μM) for 24 h. ND: undetected. **G** Viral titers in the media of the infected LECs in (**E**). **H** Viability of BC-3, BCBL-1, BJAB, and JSC-1 cells in the presence of DON. **I** Viability of BC-3, BCBL-1, BJAB, and JSC-1 cells in the presence of Palbociclib. **J** BJAB cells were infected with KSHV and selected with hygromycin (500 μg/mL). Viability of BJAB and BJAB-KSHV was determined in the presence of DON or PALA. **K** IVIS imaging of NOD-SCID mice engrafted with 2.5 ×10$^6$ BCBL-1-luciferase and treated with vehicle control or JHU083. **L** Survival curve of the 1 ×10$^7$ BC-3-luciferase-engrafted NOD-SCID mice treated with vehicle control, JHU083, or Palbociclib. Solid tumors extracted (**M**) and weighed (**N**) from the 1 ×10$^7$ JSC-1-luciferase-engrafted NOD-SCID mice treated with vehicle control, JHU083, or Palbociclib. Data are presented as mean ± SD of n = 3 biological replicates (7D-7J). Significance was calculated using two-tailed, unpaired Student's t-test. Source data are provided as a source data file. See also Figs. S7 and S8.

For plasmid transfection in HEK293T cells, cells were plated at around 60% confluence and calcium phosphate transfection method was applied. For transfection in HOKs, Lipofectamine 3000 transfection reagent (Life Technologies) was used according to the manufacturer's instructions.

### Lentivirus-mediated stable cell line construction

HEK293T cells were transfected with the packaging plasmids psPAX2 and pMD2.G, as well as a lentiviral transfer plasmid (pCDH for protein overexpression, pGIPZ or pLKO.1 for shRNA, CMV-GFP-T2A-Lucifase (System Biosciences) for luciferase expression, and pLentiGUIDE-Puro+pLentiCas9-Blast or pLentiCRISPR v2 for sgRNA). At 48 h post transfection, supernatant was harvested and filtered (and concentrated by centrifugation if necessary). HEK293T cells, TIME cell, HOKs, LECs, and PEL cells were infected with the supernatant in the presence of polybrene (0-8 μg/ml) with centrifugation at 750 g for 1 h. Cell culture medium was replaced 6 h post infection and cells were selected at 48 h post infection and maintained in medium supplemented with puromycin (1 - 2 μg/ml), hygromycin (200 - 400 μg/ml), or blasticidin (5 μg/ml).

### Cell proliferation assay

The in vitro cell viability test was performed with Cell Proliferation Kit II (XTT) according to the manufacturer's recommendation (Sigma). Briefly, TIME or PEL cells were seeded at a density of 1-2 × 10$^4$ cells per well in 96-well culture plates with increasing concentrations of DON, Palbociclib, or PALA. Cells were then cultured for 72 h and XTT was added to the medium for 2-4 h. Each well was measured by absorption at a wavelength of 450 or 475 nm, with growth medium only as the reference wavelength. Each data point was performed in triplicate. Cell viability was presented as the percentage of absorption at a given concentration of inhibitors divided by the maximum absorption at zero concentration.

### Soft-agar assay

BJAB, BC-3, BCBL-1, and JSC-1 cells were suspended in 2x growth medium (2x RPMI-1640 supplemented with 20% FBS and 1% penicillin/streptomycin) and mixed with 0.6% agarose supplemented with the indicated vehicle control or inhibitors, then plated on top of a solidified layer consisting of RPMI with 0.5% agarose in a 12-well plate. The cell layer of each well was further covered with growth medium containing indicated inhibitors. The top growth medium was changed every 3 days. After 2 weeks, the colonies were stained with Crystal Violet solution (0.005% Crystal Violet, 20% methanol in water), washed with water, and imaged by a digital camera.

### Lactate colorimetric assay

Cells were seeded and counted in 6-well or 12-well plates for the indicated hours. The medium was harvested and lactate was measured by the L-Lactate assay kit (Cayman Chemical) according to the manufacturer's instruction.

### Protein expression and purification

HEK293T cells were transfected with expression vector containing Flag-tagged genes of interest. Cells were harvested and lysed with Triton X-100 buffer (20 mM Tris, pH 7.5, 150 mM NaCl, 1.5 mM MgCl$_2$, 20 mM β-glycerophosphate, 1 mM sodium orthovanadate, 10% glycerol, 0.5 mM EGTA, 0.5% Triton X-100) supplemented with a protease inhibitor cocktail (Roche). Whole cell lysates were sonicated and centrifuged at 15,000 g for 15 min. Supernatant was harvested, filtered, pre-cleared with protein A/G agarose beads at 4 °C for 1 h and then incubated with anti-Flag agarose beads at 4 °C for 3 h. After incubation, the agarose beads were washed extensively and bound proteins were eluted with 0.2 mg/ml of 3x Flag peptide. The eluted proteins were analyzed and quantified by SDS gel electrophoresis and Coomassie blue staining.

### Co-immunoprecipitation (Co-IP) and immunoblotting

For Co-IP using ectopically expressed protein, HEK293T cells were transfected with the indicated expression plasmids for 48 h. For Co-IP using endogenous proteins, cells were directly harvested. The cell pellets were lysed in Triton X-100 buffer or NP40 buffer (50 mM Tris-HCl, pH 7.4, 150 mM NaCl, 1% NP-40, 5 mM EDTA) supplemented with 20 mM β-glycerophosphate and 1 mM sodium orthovanadate. Whole cell lysates were sonicated, centrifuged and pre-cleared with protein A/G agarose for 1 h. Pre-cleared samples were then incubated with the indicated antibodies overnight followed by protein A/G agarose for 1 h at 4 °C, or with antibody/glutathione-conjugated agarose for 3 h at 4 °C. The agarose beads were washed extensively and samples were eluted by boiling at 95 °C for 10 min. Precipitated proteins were analyzed by SDS-PAGE and immunoblotting.

All immunoblottings were performed using the indicated primary antibodies (1:500 − 1:2000 dilution), IRDye700/800-conjugated secondary antibodies (1:10,000 dilution, Licor), or Anti-mouse/rabbit IgG HRP-linked secondary antibodies (1:5000 dilution, Cell Signaling). Proteins were visualized by Odyssey infrared imaging system (Licor) or Amersham Imagequant 800.

### Dual-luciferase reporter assay

HEK293T cells, seeded in 24-well plates (~50% cell density), were transfected with NF-κB reporter plasmid cocktail (50 ng of luciferase reporter plasmid and 5 ng of pRL Renilla luciferase control vector) and the expression plasmids by calcium phosphate precipitation. Whole cell lysates at 30 h post-transfection were used to determine the activity of firefly luciferase and Renilla luciferase by a microplate reader (Synergy Neo2, BioTek). When Sendai virus (SeV) was used, cells were infected with SeV [100 HA/mL] at 24 h post-transfection and whole cell lysates were harvested at 48 h.

### CAD activity assay

CAD activity assay was modified from the CPSase activity assay described previously[34,73]. Briefly, the reaction mixture was composed of 40 μl of glutamate detection mixture (Glutamate Assay Kit, Sigma)

and 40 μl of the CAD enzymatic buffer containing ~0.4 μg of CAD, 87 mM Tris–HCl (pH 8.0), 87 mM KCl, 1.5 mM glutamine, 17.4 mM L-aspartic acid, 7 mM NaHCO₃, 2 mM MgCl₂, and 1 mM ATP. The reaction was incubated at 37 °C for 3 h in a microplate reader (Synergy Neo2, BioTek) for the kinetic assay. The absorbance was read at 565 nm in 5-minute intervals.

### In vitro deamidation assay
GST-RelA, CAD and CAD phosphorylation mutants were purified from transfected 293 T cells to homogeneity as determined by Coomassie staining. In vitro on-column deamidation of RelA was performed as previously reported[27,34]. Briefly, ~0.2 μg of CAD or CAD phosphorylation mutants, and 0.6 μg of GST-RelA (bound to glutathione-conjugated agarose) were added to a total volume of 30 μl. The reaction was carried out at 30 °C for 15 or 45 min in deamidation buffer (100 mM Tris–HCl at pH 8.0, 100 mM KCl, 1 mM dithiothreitol, 20.2 mM aspartate, 1.5 mM ATP, 200 mM phosphoribosyl pyrophosphate (PRPP), 3.5 mM MgCl₂, and 5 mM NaHCO₃). Protein-bound GST beads were washed with deamidation buffer and GST-RelA was eluted with rehydration buffer (8 M Urea, 2% CHAPS, 0.5% IPG Buffer, 0.002% bromophenol blue) at room temperature. Samples were then analyzed by two-dimensional gel electrophoresis and immunoblotting.

### In vitro kinase assay
CAD, CDK4, and CDK6 were purified from transfected 293 T cells. Reaction mixtures containing 0.2 μg of kinase (CDK4/CDK6) with or without Palbociclib, 0.5 μg of CAD, and 10 μCi of [γ³²P]-ATP in a total volume of 30 μl were incubated at 25 °C for 30 min. Reactions were stopped by adding SDS-PAGE loading buffer and boiling for 5 min at 95 °C. Samples were resolved by SDS-PAGE, transferred to nitrocellulose membrane, and analyzed by autoradiography (Typhoon).

### Two-dimensional Gel electrophoresis
Cells were lysed in 150 ml rehydration buffer [8 M Urea (or 6 M Urea/ 2 M Thiourea), 2% CHAPS, 0.5% IPG Buffer, 0.002% bromophenol blue] by one pulse of sonication and whole cell lysates were centrifuged at 18,000 g for 15 min. Supernatants were loaded to IEF strips (Cytiva) for focusing with a program comprising: 10 h (rehydration); 500 V, 1 h; 1000 V, 1 h; 1000-5000 V, 4 h; 5000 V, 3 h. After IEF, strips were incubated with SDS equilibration buffer (50 mM Tris-HCl [pH8.8], 6 M urea, 30% glycerol, 2% SDS, 0.001% Bromophenol Blue) containing 10 mg/ mL DTT for 15 min and then SDS equilibration buffer containing 2-iodoacetamide (250 mg/10 ml) for 15 min. Strips were washed with SDS-PAGE buffer, resolved by SDS-PAGE, and analyzed by immunoblotting.

### Glycolysis rate assay (seahorse)
2 × 10⁴ of HOK16B cells were seeded on a XFp cell culture plate. 24 h later the cells were then infected with KSHV (MOI = 30) for 24 h. For HOK16B stable cell line expressing empty vector or vCyclin, 3.5 ×10⁴ cells were seeded on the XFp plate for 18 h. Real time extracellular acidification rate (ECAR) and oxygen consumption rate (OCR) was measured using the glycolytic rate assay kit in Seahorse XF Base Medium (10 mM glucose, 2 mM L-glutamine, and 1 mM sodium pyruvate) under basal conditions and in response to 0.5 μM rotenone/ antimycin A and 50 mM 2-Deoxy-D-glucose with the Seahorse XF HS mini Analyzer (Agilent). Cell numbers were counted after the assay and used to normalize ECAR and OCR.

### Metabolomics and isotope tracing
For extraction of intracellular metabolites, cells were washed with 1 mL ice-cold 150 mM ammonium acetate (NH₄AcO, pH 7.3). After that, 1 mL of −80 °C cold 80% MeOH was added to the wells, and samples were incubated at −80 °C for 20 min before cells were scraped off and transferred into tubes and centrifuged at 4 °C for 10 min at 21,000 g.

The supernatants were transferred into new tubes, and the cell pellets were re-extracted with 200 μl ice-cold 80% MeOH, spun down and the supernatants were combined. Metabolites were dried at room temperature under vacuum and re-suspended in water for LC-MS run. Targeted Metabolomics analyses were performed at Proteomics and Metabolomics Core of Cleveland Clinic Lerner Research Institute.

Isotope tracing experiments were performed as previously described[74–76]. For glycolysis tracing, cells were cultured with medium containing [U-¹³C]-labeled glucose for 15 min. For de novo pyrimidine synthesis activity, cells were cultured with medium containing [Amide-¹⁵N] glutamine for 15 min. Metabolite extraction was then performed.

Samples were randomized and analyzed on a Q-Exactive Plus hybrid quadrupole-Orbitrap mass spectrometer coupled to Vanquish UHPLC system (Thermo Fisher). The mass spectrometer was run in polarity switching mode (+3.00 kV/−2.25 kV) with an m/z window ranging from 65 to 975. Mobile phase A was 5 mM NH₄AcO, pH 9.9, and mobile phase B was acetonitrile. Metabolites were separated on a Luna 3 μm NH₂ 100 Å (150 × 2.0 mm) column (Phenomenex). The flow rate was 300 μl/min, and the gradient was from 15% A to 95% A in 18 min, followed by an isocratic step for 9 min and re-equilibration for 7 min. All samples were run in at least three biological replicates.

Metabolites were detected and quantified as area under the curve based on retention time and accurate mass (≤ 5 ppm) using Trace-Finder 4.1 (Thermo Scientific) software against known external standards. Raw data was corrected for naturally occurring ¹³C and ¹⁵N abundance and tracer impurity using the IsoCorrectoR package[77]. The full panel of metabolites were then subjected to cell number normalization and data were presented as normalized peak area. Metabolite levels were further compared using a two-tailed, unpaired Student's t test.

Full name of the metabolites. PEP: phosphoenolpyruvate; 2-PG: 2-phosphoglyceric acid; 3-PG: 3-phosphoglyceric acid; G-6-P: glucose-6-phosphate; F-6-P: fructose-6-phosphate; F-1,6-BP: fructose-1,6-biphosphate; 1,3-BPG: 1,3-biphosphoglyceric acid; DHAP: dihydroxyacetone phosphate; GADP: glyceraldehyde-3-phosphate; NAD: nicotinamide adenine dinucleotide; NAM: nicotinamide; NMN: nicotinamide mononucleotide; UMP: uridine monophosphate; UDP: uridine diphosphate; IMP: inosine monophosphate; AMP: adenosine monophosphate; Ribulose-1,5-BP: ribulose-1,5-biphosphoate; 6-PG: 6-phosphogluconate; Ribose-5-P: ribose-5-phosphoate; α-KG: alpha-ketoglutaric acid.

### Mass spectrometry analysis
To identify CAD phosphorylation site(s) by vCyclin and CDK6, HEK293T cells were transfected with an expression plasmid containing CDK6 or vCyclin. Whole cell lysates (WCLs) were prepared with the lysis buffer (8 M Urea, 100 mM Tris, 50 mM β-glycerophosphate, and 1 mM sodium orthovanadate) and subjected to sonication, reduction (5 mM DTT), and alkylation (25 mM IAA). WCLs were then digested with trypsin overnight and peptides were purified by the C18 Spin Tips (Thermofisher) according to the manufacturer's instruction, before the samples were dried by Speedvac centrifugation (Thermofisher).

Phosphopeptide enrichment was performed as previously described[78]. Briefly, the tryptic peptides were resuspended in loading buffer (80% CAN, 6% TFA) and transferred to the Titanium Dioxide (TiO₂) spin columns (Thermofisher). The columns were then washed extensively with the wash buffer (50% ACN, 0.1% TFA) and phosphopeptides were eluted with the elution buffers (1: 10% NH₄OH, 2: 80% ACN, 2% FA). Finally, samples were dried at room temperature by Speedvac centrifugation and re-suspended in water containing 0.1% FA for further LC-MS/MS analysis.

LC-MS/MS analysis was performed with a Dionex UltiMate 3000 HPLC system (ThermoFisher Scientific) coupled to a Q-Exactive Plus hybrid quadrupole-Orbitrap mass spectrometer (ThermoFisher

Scientific). Peptides were separated on the heated EASY-Spray analytical column (C18, 2 μm, 100 Å, 75 μm × 25 cm, ThermoFisher Scientific) with a flow rate of 0.3 μL/min for a total duration of 147 min and ionized at 2.0 kV in the positive ion mode. The gradient was composed of 3-38% buffer B (132 min) followed by the wash step at 98% B (15 min); solvent A: 0.1% FA; solvent B: 80% ACN and 0.1% FA. MS1 scans for data-dependent acquisition were acquired at the resolution of 70,000 from 350 to 1800 m/z, AGC target 1e6, and maximum injection time 100 ms. The 10 most abundant ions in MS2 scans were acquired at a resolution of 17,500, AGC target 5e4, maximum injection time 120 ms, and normalized collision energy of 28. Dynamic exclusion was set to 30 s and ions with charge +1, +7 and >+7 were excluded. MS2 fragmentation spectra were searched with Proteome Discoverer SEQUEST (version 2.4, Thermo Scientific) against in silico tryptic digested Uniprot all-reviewed Homo sapiens database. The maximum missed cleavages was set to 2. Dynamic modifications were set to oxidation on methionine (M, +15.995 Da), phosphorylation on serine, threonine, or tyrosine (S, T, and Y, +79.966 Da). Carbamidomethylation on cysteine (C, +57.021 Da) was set as a fixed modification. The maximum parental mass error was set to 10 ppm, and the MS2 mass tolerance was set to 0.02 Da. The false discovery threshold was set strictly to 0.01 using the Percolator Node validated by q-value. Phosphosite localization probabilities were determined by the IMP-ptmRS node, and only phosphosites with >0.75% localization probability were selected. The relative abundance of parental peptides was calculated by integration of the area under the curve of the MS1 peaks using the Minora LFQ node. Spectral annotation was generated by the Interactive Peptide Spectral Annotator (IPSA)[79].

### Real-time quantitative PCR (RT-qPCR)

Real-time Quantitative PCR was performed as previously described[34]. Total RNA was extracted using TRIzol reagent (Invitrogen). Complementary cDNA was synthesized from DNase I-treated total RNA using reverse transcriptase (Takara). cDNA was diluted and RT-qPCR was performed using SYBR Green Master Mix (Bio-Rad and Genesee) by real-time PCR instrument (Bio-Rad CFX). Relative mRNA expression for each target gene was calculated by the $2^{-\Delta\Delta Ct}$ method using *β-actin* as an internal control. The qPCR results were presented as fold changes relative to the control or mock group. The sequences of RT-qPCR primers are listed in Supplementary Table 1.

### Statistical analysis

Data are presented as mean ± standard deviation (SD), if not stated in the figure legend. Statistical analyses were performed by un-paired, two-tailed Student's t-test (unless specified). Statistical analyses were performed with Excel or Graphpad Prism. A p value less than 0.05 is considered statistically significant. Power analysis was used to estimate the sample size for in vivo experiments.

### Reporting summary

Further information on research design is available in the Nature Portfolio Reporting Summary linked to this article.

## Data availability

The mass spectrometry proteomics data have been deposited to the ProteomeXchange[80] Consortium via the PRIDE[81] partner repository with the dataset identifier PXD043435. All data needed to evaluate the conclusions for the study are present in the paper. Unique expression vectors and reagents generated for this study are available upon request. Source data are provided with this paper.

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

## Acknowledgements
We thank Dr. Pinghui Feng (University of Southern California) for cell lines and the support in mass spectrometry analysis. We are also grateful to Dr. Jae U Jung (Cleveland Clinic), Dr. Michael Lagunoff (University of Washington), Dr. Ren Sun (University of California Los Angeles) and Dr. Young-Kwon Hong (University of Southern California) for cell lines and KSHV reagents. We also thank Cleveland Clinic Lerner Research Institute Proteomics and Metabolomics Core, as well as Florida Research and Innovation Flow Cytometry Core for services and analyses. J.Z. is supported by a grant from NIDCR (R00DE028973), startup funds from the Cleveland Clinic Florida Research and Innovation Center, and a Cleveland Clinic Global Center for Pathogen and Human Health Research Fast Track Research Award. Z.H. is supported by a grant from NIAID (R01AI173277).

## Author contributions
Q.W. and J.Z. designed the experiments. Q.W., L.T., T.Y.W., A.J.T., R.Z., and J.Z. performed the experiments. Q.L., S.F. and D.C. contributed to key biological resources. Q.W., L.T., T.Y.W., A.J.T., R.Z., Z.H., M.U.G. and J.Z. contributed to the interpretation of the data. J.Z. conceived the study. Q.W., M.U.G. and J.Z. wrote the manuscript, with input from all the authors.

## Competing interests
The authors declare no competing interests.
