## [Peer Review File · Nature Communications]

Hijacking of nucleotide biosynthesis and deamidation-mediated glycolysis by an oncogenic herpesvirusREVIEWER COMMENTS

Reviewer #1 (Remarks to the Author):

In this manuscript, Wan et al. explore the mechanisms of metabolic reprogramming mediated by KSHV. Their findings reveal that KSHV infection enhances CAD activity and induces RelA deamidation, thereby modulating pyrimidine synthesis and glycolysis via the interaction of KSHV vCyclin with CAD. They further demonstrate that KSHV vCyclin associates with CDK6, leading to the phosphorylation of CAD S1900 and the activation of CAD functions. Leveraging these insights, the authors show that the pharmacological inhibition of CDK6 and CAD robustly suppresses tumorigenesis in PEL cells NOD-SCID mice and hampers KSHV lytic replication.

Overall, while the study appears to be thoughtfully designed and appropriately conducted, the findings, although potentially significant, and the proposed treatment model do require further substantiation. There are several aspects that need clarification or more detailed exploration.

1. The credibility of the conclusions could be impacted by the observed inconsistencies in sampling times and target genes tested in Figure 1D, E, G, and H. Please ensure consistency in grouping methods, such as uniformly sampling time points and presenting the same group of genes tested with a standardized format to ensure consistency and clarity.
2. The interaction between vCyclin and CAD was only examined in HEK293 cells, which somewhat limits the conclusiveness of the results. Verifying this interaction in KSHV-infected cells or employing other methods could potentially strengthen this conclusion.
3. When comparing with Mock, there is approximately a 50-fold change of Dihydroorotic acid, the downstream metabolite directly regulated by CAD, a 150-fold change of indirectly regulated metabolite Orotic acid, but only a 3 to 4-fold change of final product cytidine triphosphate. Please provide further commentary on the metabolic isotope tracing results in Figure 3F.
4. Figures 3G-I lack clarity in illustrating how vCyclin promotes glycolysis and pyrimidine synthesis, and its necessity for KSHV proliferation. This could be addressed by adding a rescue with metabolic substrates to check whether viral mRNA expression and KSHV replication can be restored in Δ Cyclin infected cells.
5. In Figure 6, the authors state that CAD is essential for optimal growth of another PEL cell line, BC-3 (Figures S6K and S6L), a finding that contrasts significantly with its seemingly marginal role in supporting 293T proliferation in vitro (Figure S6M). The difference in CAD knockout effects can be due to many reasons. Either you demonstrate that is specific to KSHV-positive cells, (provide the CAD dependency information in KSHV-infected and non-infected cells) or better I suggest not including this data as it can be confusing.
6. In figure 6k, a rescue group expressing RelA mutant DD would be helpful to verify the functions of the CAD-RelA pathway in KSHV-infected cells.
7. Regarding the use of CDK4/6 inhibitors, prior studies show PEL cell lines exhibit high sensitivity to Palbociclib (Manzano et al., Nature Communications, 2018). Also, Wu et al. (Journal of Translational Medicine, 2022) found that CDK4/6 inhibitors, such as abemaciclib, palbociclib, and ribociclib, inhibited cell growth in KSHV-induced primary effusion lymphoma (PEL) and EBV positive Burkitt's lymphoma (BL) cell lines, and in KSHV-infected human umbilical vein endothelial cells (HUVECs). Hence, it's not surprising that CDK6 inhibitor palbociclib blocks cell growth and tumorigenesis in BCBL-1 and JSC-1 in Figure 7. If Palbociclib blocks KSHV viral pathogenesis by inhibiting CAD's functions (as suggested by the mechanisms outlined in this manuscript), one would expect to see a rescue following the addition of metabolites or the introduction of RelA DD. Additionally, Figure 7E should also include treatment with DON or Palbociclib without KSHV infection to ascertain whether DON or Palbociclib directly influences cell metabolism and lactate concentration.

Minor:

1. For maintaining uniformity in the use of exponential notation across your paper, it would be advantageous to adhere to a single format as demonstrated in Figure 1H, I and Figure 6H-J.
2. Figure 1F, S3G, and S3F lack clear identification of band changes without quantification. This is observed for the PDK3 band in Figure 1F, the PC and HK4 bands in Figure S3G, and the PDK3 band in Figure S3F.
3. The methodologies used for Seahorse are missing.

Reviewer #2 (Remarks to the Author):

In this very interesting Wan and colleagues report the novel observation that the KSHV viral cyclin homologue activates the CAD multifunctional enzyme by mediating, in complex with cellular cdk6, its phosphorylation on S1900, and thereby activating its ability to deamidate RelA and stimulate CAD-dependent metabolic changes in the KSHV-infected cell. Furthermore, the authors attempt to show that pharmacological targeting of CAD could be used to inhibit the growth of KSHV-infected PEL cells in tissue culture and in an in vivo lymphoma model.

The experimental evidence provided in this manuscript to support these conclusions is very comprehensive and also largely very convincing. There are a few points, where alternative interpretations of their findings might also be possible (see below). Furthermore, their attempts to show that pharmacological targeting of CAD could be a way to specifically inhibit the growth of KSHV-infected PEL tumours probably fall short of making this point convincingly, as also explained below. These points should be addressed before the manuscript is accepted for publication.

Major points needing to be addressed experimentally:

1. Fig. 3: How were equal infection levels of KSHVwt and KSHVdelvcyc verified? In fig. S3E and S3F, LANA levels in KSHVwt-infected TIME and HOK cells are higher than in KSHVdel vcyc-infected cells.
2. Fig.3: the effects of a lack of vcyc on glycolysis, viral gene expression and viral progeny production shown in this figure could be indirect, i.e. vcyc could affect the lytic cycle and metabolic changes might be secondary to this and independent of the effect of vcyc on RelA deamidation. It would therefore be necessary to show similar metabolic changes by overexpressing vcyc. Regarding lactate production, this has been done (figure 2I, J) but investigating if overexpression of vcyc induces similar metabolic changes as seen in KSHVwt- vs. KSHV-del vcyc – infected cells would be important.
3. Fig. S4C: also in this case, the effects induced by knocking down CDK6 could be indirect: as knocking down cdk6 inhibits lytic replication, it is the absence of lytic replication that could cause the changes in the expression of KK4, PDK3, etc. shown in this figure.
4. Fig. 5E: Here CAD S1900A still seems to enhance RelA deamidation in comparison to the control sample, perhaps suggesting that phosphorylation of CAD on S1900 is not the only mechanism how vcyc/cdk6 activate CAD.
5. Fig. 5H: Similarly, CAD1900A still seems to enhance virus production in comparison to the control sample, also perhaps suggesting that phosphorylation of CAD on S1900 is not the only mechanism how vcyc/cdk6 activate CAD.
6. Fig. S5B: there is still a reduction in NFkB activity when CAD S1900A is overexpressed, the effect of CAD S1900A is similar to that of CAD WT, again that phosphorylation of CAD on S1900 is not the only mechanism how vcyc/cdk6 activate CAD.
7. Figure S7B: PALA does not inhibit BCBL-1 any better than BJAB, suggesting that its inhibition of CAD ATCase activity is not specific for KSHV-infected cells.
8. Figure 7H: DON inhibits PEL cell lines as efficiently as (BC3) or better than (JSC1) BJAB, again suggesting that inhibiting CAD activity does not exert a preferential effect on KSHV-infected cells and KSHV-dependent proliferation. This would argue against one of the key arguments of this manuscript, i.e that the metabolic changes due to the activation of CAD by vcyc/cdk6 could be exploited to specifically/preferentially inhibit the growth of KSHV-infected tumor cells.
9. Figure S7 C-E: it would be important to show the effect of DON, PALA and Palbo on PEL cells in the colony forming assay in comparison to their effect on BJAB.

Minor points:

1. Lines 178, 187, 197, 198, 229, 497: name of KSHV with a vcyc stop mutation not clear
2. Line 265, 267, 358: NFkB spelling incorrect
3. Line 510: cells were spun, not 'spinned'

Reviewer #3 (Remarks to the Author):

The submitted manuscript, "Hijacking of Nucleotide Biosynthesis and Deamidation-mediated Glycolysis by an Oncogenic Herpesvirus" by Wan et al., is a continuation of investigations focused on the role of pyrimidine synthesis, CAD deamidation activity, and glycolysis. The authors previously found that (CAD) deamidates the RelA subunit of NF- κ B to promote aerobic glycolysis and promote proliferation of cancer cells. Now they are applying that information to KSHV infection. The authors are proposing that KSHV vCyclin activates CDK6 to phosphorylate CAD leading to increased aerobic glycolysis and cell proliferation through deamidation of RelA. The study has several noteworthy observations (e.g., CAD phosphorylation at S1900 by CDK6 is activating, the necessity of CAD in KSHV tumorigenesis using an in vivo mouse model, and that vCyclin impacts metabolism in a CAD-dependent mechanism). The experiments use a variety of approaches and tools (several cells, in vivo, and in vitro assays). Some conclusions are drawn from well performed experiments, such as Fig 2I-J. Overall, the manuscript reports a novel, interesting concept; however, there are several opportunities for improvement and a need to support the claims that are made. Much of these comments are addressable by either inclusion of additional experiments or altering the manuscript to limit conclusions to what is possible with the current experiments.

1. One of the major conclusions—i.e., that the effect of CAD on glycolysis in KSHV infection is a "RelA-deamidation-mediated glycolytic reprogramming" mechanism—is not directly shown and depends on evidence that is correlative. This conclusion appears to mostly be dependent on Fig 6H, which is not overly convincing by itself. It may be possible (likely?) that CAD is impacting glycolysis through a RelA-independent mechanism and that the role of RelA may be, at the most, a minor contributor.
2. The conclusions depend on many blots; however, no quantitative information is provided for the blots or through a complementing method.
3. Moreover, it is not clear if the blots represent several independent biological replicates.
4. Many claims regarding glycolysis and 'metabolism' are based only on lactate measurement, which may be insufficient to draw strong conclusions regarding the whole glycolytic pathway or a broader picture of the metabolic network.
5. Similar to point #4, de novo pyrimidine synthesis is a major focus of conclusions; however, measurements of pyrimidines are rarely shown.
6. There does not appear to be a strong correlation between WT, S1900D, and S1900A regarding metabolism and virus replication. E.g., WT and S1900A have similar levels of dihydroorotic acid and lactate, but it appears that S1900A has a defect in virus replication as measured by PFU. This weakens the overall claim. Is there a reason for this observation?
7. The manuscript lacks rationale, explanation or justification for the many different cells, time points, MOIs, metabolic genes examined, etc. reducing the ability of the work to be appreciated by a general audience.
8. The conclusion that "KSHV vCyclin promotes glycolysis and de novo pyrimidine synthesis, and is required for optimal viral replication" is not demonstrated by the experimental evidence. Does restoring glycolysis and de novo pyrimidine synthesis restore virus replication in the vCyclin mutant virus?
9. Many experiments depend on genetic or small molecule/inhibitor treatment, however their effect on cell health/survivability is not included when interpreting the results. For example, infection with the vCyclin mutant virus causes a decrease in all the reported metabolites (Fig 3B). However, this may not be specific to glycolysis or pyrimidine synthesis as the authors conclude. Perhaps all metabolites are decreased due to differences in infection rate or the ability of the virus to replicate, or even the cell health, survivability.

Minor items to address:

1. Since the labeling data is provided for only a single 15 min time point, it is not possible to determine a rate and thus the usage of metabolic 'flux' and 'rate analysis' is not appropriate.
2. Several data are presented as "relative" or "normalized" however it is not always clear what is meant by these terms.
3. The image resolution of Fig 4E is too poor, the image is unreadable.

General Response:

We thank all three reviewers for their critical and constructive comments on our manuscript. Although all reviewers appreciated the significance and comprehensiveness of our study, they also raised excellent questions and provided insightful comments. To this end, we have strived to address the questions raised by the reviewers by performing additional experiments. For those that are not addressed experimentally, we have provided clarifications. We hope that the reviewers will appreciate the significant improvement of our revised manuscript.

Reviewer #1 (Remarks to the Author):

In this manuscript, Wan et al. explore the mechanisms of metabolic reprogramming mediated by KSHV. Their findings reveal that KSHV infection enhances CAD activity and induces RelA deamidation, thereby modulating pyrimidine synthesis and glycolysis via the interaction of KSHV vCyclin with CAD. They further demonstrate that KSHV vCyclin associates with CDK6, leading to the phosphorylation of CAD S1900 and the activation of CAD functions. Leveraging these insights, the authors show that the pharmacological inhibition of CDK6 and CAD robustly suppresses tumorigenesis in PEL cells NOD-SCID mice and hampers KSHV lytic replication.

Overall, while the study appears to be thoughtfully designed and appropriately conducted, the findings, although potentially significant, and the proposed treatment model do require further substantiation. There are several aspects that need clarification or more detailed exploration.

Response: We thank the reviewer for his/her positive evaluation of our study, in particular its significance and our approaches. The reviewer raised important questions, which we have now addressed experimentally. Please see our detailed point-by-point response below.

1. The credibility of the conclusions could be impacted by the observed inconsistencies in sampling times and target genes tested in Figure 1D, E, G, and H. Please ensure consistency in grouping methods, such as uniformly sampling time points and presenting the same group of genes tested with a standardized format to ensure consistency and clarity.

Response: Thank you for the excellent suggestion. We have now repeated the real-time PCR and the western blotting experiments for both KSHV-infected HOK and TIME cells. We have now chosen the same target genes (*HK4*, *PC*, and *PDK3*) and included multiple time points: 24-, 48-, and 72-hours post infection for qPCR; 72-hours post infection for western blotting. The data support our conclusion that *de novo* KSHV infection promotes RelA deamidation and deamidation-mediated metabolic gene expression.

Figure 1D

Figure 1E

Figure 1F and 1G

2. The interaction between vCyclin and CAD was only examined in HEK293 cells, which somewhat limits the conclusiveness of the results. Verifying this interaction in KSHV-infected cells or employing other methods could potentially strengthen this conclusion.

Response: Thank you for this suggestion. We have now validated the interaction between virally expressed vCyclin and endogenous CAD in the KSHV-positive lymphoma cell line BC-3 (new Figure S2C).

Figure S2C

3. When comparing with Mock, there is approximately a 50-fold change of Dihydroorotic acid, the downstream metabolite directly regulated by CAD, a 150-fold change of indirectly regulated metabolite Orotic acid, but only a 3 to 4-fold change of final product cytidine triphosphate. Please provide further commentary on the metabolic isotope tracing results in Figure 3F.

Response: Thank you for raising this point. The levels of the isotope-incorporated metabolites were calculated and presented based on peak areas divided by the cell number. Thus, the overall signals of ¹⁵N-labeled dihydroorotic acid and orotic acid were significantly higher than CTP. One of the explanations is that we only incubated the cells with ¹⁵N glutamine for 15 min to enrich ¹⁵N-labeled CAD reaction product, thereby defining the effect of KSHV on the activity of CAD. During this short time, the majority of the ¹⁵N has not reached to the end product of pyrimidine pathway: CTP, which likely diminished its differences between mock, WT, and ΔCyclin-infected cells. Another possible reason is that negative feedback loops exist in these sequential steps of the de novo pyrimidine biosynthesis to balance the final nucleotide pool after abrupt upstream changes. Extensive additional experimentation and optimization will be needed to delineate the nucleotide pool changes in KSHV-infected cells, which we believe is outside the scope of the current manuscript.

4. Figures 3G-I lack clarity in illustrating how vCyclin promotes glycolysis and pyrimidine synthesis, and its necessity for KSHV proliferation. This could be addressed by adding a rescue with metabolic substrates to check whether viral mRNA expression and KSHV replication can be restored in ΔCyclin infected cells.

Response: The reviewer's point is well-taken. We have now performed *de novo* infection with ΔCyclin in uridine-supplemented HOKs. Uridine can be converted to UMP, UTP, CMP, etc. via the pyrimidine salvage pathway, thus compensating for the *de novo* pyrimidine biosynthesis. Interestingly, uridine did not restore but modestly inhibited viral mRNA expression (new Figures 3H (replacing the original 3H) and S3K), which can be explained by the fact that the uridine-derived UTP is a negative regulator of CAD (Tatibana and Shigesada, 1972). Thus, uridine addition may block CAD-mediated RelA deamidation and thereby impact RelA-DD-mediated viral gene expression (as observed in Figure 6A). Nevertheless, the contribution of uridine to nucleotide surpassed its negative effect on viral mRNA expression, resulting in an overall increase of viral titers (Improved Figure 3I). Notably, even with uridine addition, the viral titers of ΔCyclin failed to reach the titers of WT virus.

Figure 3H**Figure S3K****Figure 3I**
To further induce deamidation-mediated glycolysis and viral gene expression, we depleted HOK cells of endogenous RelA with shRNA targeting its 3'UTR and then reconstituted these cells with RelA-DD. We found that uridine addition and RelA-DD expression further restored the replication of ΔCyclin (new Figure S3L), demonstrating the pivotal role of vCyclin in driving deamidation and pyrimidine biosynthesis. Notably, RelA-DD reconstitution and uridine addition failed to restore the viral titers of ΔCyclin to those of WT virus, when compared to the positive control of vCyclin overexpression (new Figure S3M). This indicates that vCyclin may have additional mechanisms to promote viral lytic replication, which are currently being explored in our lab. We have added discussion of the result (page 9, line 443-447).

Figure S3L**Figure S3M**
5. In Figure 6, the authors state that CAD is essential for optimal growth of another PEL cell line, BC-3 (Figures S6K and S6L), a finding that contrasts significantly with its seemingly marginal role in supporting 293T proliferation in vitro (Figure S6M). The difference in CAD knockout effects can be due to many reasons. Either you demonstrate that is specific to KSHV-positive cells, (provide the CAD dependency information in KSHV-infected and non-infected cells) or better I suggest not including this data as it can be confusing.

Response: We thank the reviewer for this thoughtful and constructive suggestion. We agree with the reviewer that comparing CAD depletion effects for different cell lines is not a strong evidence to support CAD dependency for KSHV-positive cells. Therefore, we have removed the 293T data from the manuscript. Notably, by applying CAD inhibitors PALA and DON, we did observe a more potent effect on cell viability for KSHV-infected TIME cells (Figure 6K) and BJAB cells (new Figure 7J) compared to the mock-infected control cells at lower concentration. Thus, our preliminary data suggest that CAD is a potential metabolic vulnerability of KSHV-infected cells.

Figure 6K

Figure 7J

6. In figure 6k, a rescue group expressing RelA mutant DD would be helpful to verify the functions of the CAD-RelA pathway in KSHV-infected cells.

Response: Thank you for this excellent suggestion. We have now reconstituted BCBL-1 cells with RelA-DD via lentiviral transduction, followed by CAD depletion and performing XTT proliferation assays. We found that RelA-DD expression marginally promoted the proliferation of BC-3 cells, which is likely due to the presence of the endogenous deamidated RelA. However, RelA-DD reconstitution partially restored the proliferation of BCBL-1 cells upon CAD depletion (new Figure S6J). Altogether, this data suggests that the contribution of CAD to BCBL-1 survival and propagation is in part mediated by RelA deamidation.

Figure S6J

7. Regarding the use of CDK4/6 inhibitors, prior studies show PEL cell lines exhibit high sensitivity to Palbociclib (Manzano et al., Nature Communications, 2018). Also, Wu et al. (Journal of Translational Medicine, 2022) found that CDK4/6 inhibitors, such as abemaciclib, palbociclib, and ribociclib, inhibited cell growth in KSHV-induced primary effusion lymphoma (PEL) and EBV positive Burkitt's lymphoma (BL)

cell lines, and in KSHV-infected human umbilical vein endothelial cells (HUVECs). Hence, it's not surprising that CDK6 inhibitor palbociclib blocks cell growth and tumorigenesis in BCBL-1 and JSC-1 in Figure 7. If Palbociclib blocks KSHV viral pathogenesis by inhibiting CAD's functions (as suggested by the mechanisms outlined in this manuscript), one would expect to see a rescue following the addition of metabolites or the introduction of RelA DD. Additionally, Figure 7E should also include treatment with DON or Palbociclib without KSHV infection to ascertain whether DON or Palbociclib directly influences cell metabolism and lactate concentration.

Response: We thank the reviewer for pointing out the previous findings that corroborate our results. Some of the citations were accidentally missed during the original submission and we sincerely apologize for that. We have now cited these papers (Hollingworth et al., 2020; Manzano et al., 2018; Wu et al., 2022) in the appropriate places. We also would like to point out that our study for the first time assessed the *in vivo* efficacy of Palbociclib for primary effusion lymphoma cells, further supporting the repurposing of the FDA-approved drug for KSHV/EBV-associated malignancies.

As requested by the reviewer, we have stably reconstituted BC-3 cells with RelA-DD and treated these cells with Palbociclib. We found that indeed BC-3-DD cells showed resistance to Palbociclib, whereas addition of uridine further conferred drug resistance (new Figure S7E). This data shows that besides cell-cycle progression, the metabolic functions of CDK6 in BC-3 cell plays important roles in supporting cell proliferation.

We have also treated the cells with DON and Palbociclib in the absence of KSHV infection. Treatment of both molecules reduced lactate secretion (new Figure S7A), albeit with a lesser effect compared to KSHV-infected cells. This also suggests that cellular cyclin D(s) – CDK4/6 may regulate CAD activity to induce a basal level of deamidation and glycolysis in LECs, which we have now discussed in the Discussion section (page 8, line 398-400).

Figure S7E

Figure S7A

Minor:

1. For maintaining uniformity in the use of exponential notation across your paper, it would be advantageous to adhere to a single format as demonstrated in Figure 1H, I and Figure 6H-J.

Response: Thank you for your suggestion. We have revised the notations to keep consistency throughout the manuscript.

2. Figure 1F, S3G, and S3F lack clear identification of band changes without quantification. This is observed for the PDK3 band in Figure 1F, the PC and HK4 bands in Figure S3G, and the PDK3 band in Figure S3F.

Response: Thank you for the excellent point. As is also suggested by reviewer 3, we have repeated Figure 1F and 1G. Due to poor signals for PC and non-specific bands for HK4 in the original Figure S3G, we have also repeated it with new antibodies. We have also provided quantifications for Figure 1F, 1G, S3E-S3G. In summary, KSHV-mediated RelA deamidation induced the expression of all three genes, albeit to varying levels in the different cell types. In particular for PDK3 in KSHV-infected HOK cells, the upregulation was marginal for 72 h (Figure 1F) and not observed for 48 h (Figure S3F), which may be due to its high endogenous level in HOK cells regardless of infection.

3. The methodologies used for Seahorse are missing.

Response: We have now added the details describing the glycolysis rate assay (Seahorse) to the Methods section.

Reviewer #2 (Remarks to the Author):

In this very interesting Wan and colleagues report the novel observation that the KSHV viral cyclin homologue activates the CAD multifunctional enzyme by mediating, in complex with cellular cdk6, its phosphorylation on S1900, and thereby activating its ability to deamidate RelA and stimulate CAD-dependent metabolic changes in the KSHV-infected cell. Furthermore, the authors attempt to show that pharmacological targeting of CAD could be used to inhibit the growth of KSHV-infected PEL cells in tissue culture and in an in vivo lymphoma model.

The experimental evidence provided in this manuscript to support these conclusions is very comprehensive and also largely very convincing. There are a few points, where alternative interpretations of their findings might also be possible (see below). Furthermore, their attempts to show that pharmacological targeting of CAD could be a way to specifically inhibit the growth of KSHV-infected PEL tumours probably fall short of making this point convincingly, as also explained below. These points should be addressed before the manuscript is accepted for publication.

Response: We are thrilled to see that the reviewer appreciated the comprehensiveness of our experimental approaches. We have performed additional experiments and added new data supporting the claim that CAD serves as a metabolic vulnerability for KSHV-infected cells. Please see our detailed point-by-point response below.

Major points needing to be addressed experimentally:

1. Fig. 3: How were equal infection levels of KSHVwt and KSHVdelvcyc verified? In fig. S3E and S3F, LANA levels in KSHVwt-infected TIME and HOK cells are higher than in KSHVdelvcyc-infected cells.

Response: Thank you for raising this point. The viral titers of WT and Δ Cyclin were determined by counting total GFP+ cells upon viral serial dilutions and infections in 293T/HOK cells (please see details in the Methods section). We have provided quantification for the bands in S3E, S3F, and S3G. Due to the overall low signals for LANA in S3E, we have replaced it with a new figure using the same whole cell lysates. In response to reviewer 1, we have also repeated the whole experiment in Fig. S3G. In summary, we observed that 48 hours post infection LANA protein levels in LECs were comparable between WT and Δ Cyclin. However in HOK cells Δ Cyclin showed a slight reduction. The reduction of LANA expression was much more drastically in TIME cells. We think that the reduced expression of LANA (particularly in KSHV-infected TIME cell) can be explained by the lack of vCyclin-induced RelA deamidation and deamidation-mediated LANA expression as shown by Figure 6C.

2. Fig.3: the effects of a lack of vcyc on glycolysis, viral gene expression and viral progeny production shown in this figure could be indirect, i.e. vcyc could affect the lytic cycle and metabolic changes might be secondary to this and independent of the effect of vcyc on RelA deamidation. It would therefore be necessary to show similar metabolic changes by overexpressing vcyc. Regarding lactate production, this has been done (figure 2I, J) but investigating if overexpression of vcyc induces similar metabolic changes as seen in KSHVwt- vs. KSHV-delvcyc – infected cells would be important.

Response: We thank the reviewer for this excellent point. We have now assessed the role of vCyclin on glycolysis and CAD activation in the absence of viral lytic cycle. Specifically, we performed a Seahorse assay on vCyclin-expressing HOK stable cells and observed that vCyclin alone was sufficient to drive glycolysis (new Figure 2K). Notably, the effect of vCyclin on glycolysis was weaker than that observed during KSHV infection, indicating that vCyclin-independent mechanisms, as reported by other groups (Cai et al., 2006; Ma et al., 2015; Yogev et al., 2014), were driving glycolysis concomitantly.

Figure 2K

In addition to glycolysis, we have also pulled down endogenous CAD and probed for S1900 phosphorylation in vCyclin-expressing cells (new Figure S4G). We observed that vCyclin expression induced CAD S1900 phosphorylation, which was ablated after CDK6 knockdown.

Figure S4G

These new data suggest that overexpression of vCyclin is capable of inducing glycolysis and CAD activation, further complementing the results of KSHV-WT and Δ Cyclin-infected cells.

3. Fig. S4C: also in this case, the effects induced by knocking down CDK6 could be indirect: as knocking down cdk6 inhibits lytic replication, it is the absence of lytic replication that could cause the changes in the expression of KK4, PDK3, etc. shown in this figure.

Response: The reviewer's point is well taken. To exclude the potential effect of KSHV lytic replication, we again expressed vCyclin in HOK cells by lentiviral transduction. We found that vCyclin expression was sufficient to drive RelA deamidation-dependent metabolic gene *HK4* expression. However, upon CDK6 depletion the effect was abolished (new Figure S4H). This data further support that CDK6 is required for vCyclin to reprogram glycolysis regardless of viral lytic replication.

Figure S4H

4. Fig. 5E: Here CAD S1900A still seems to enhance RelA deamidation in comparison to the control sample, perhaps suggesting that phosphorylation of CAD on S1900 is not the only mechanism how vCyclin/cdk6 activate CAD.

Response: We thank the reviewer for this comment. We would like to point out that S1900A is not a dead enzyme. In our *in vitro* enzymatic assay it showed a reduction compared to WT but still possessed

glutaminase activity (Figure 5B). In cells S1900A behaved similarly to WT (likely due to overexpression in 293T cells) in catalyzing the formation of dihydroorotate, while S1900D showed significantly enhanced activity. We have now revised our conclusion by clarifying that S1900 promotes CAD activity but does not serve as an on-off switch for CAD enzymatic functions (page 9, line 410-412).

For Fig.5E, it is thus not surprising that overexpression of S1900A also deamidated RelA. We have re-tested the expression of the CAD mutants for the original data and found that the levels were not balanced. We thus remade the cells with similar CAD expression levels and repeated the 2DGE experiments. S1900A still induced RelA deamidation but the effect was dampened compared to WT and S1900D (new Figure 5E).

Figure 5B

Figure 5E

5. Fig. 5H: Similarly, CAD1900A still seems to enhance virus production in comparison to the control sample, also perhaps suggesting that phosphorylation of CAD on S1900 is not the only mechanism how vCyc/cdk6 activate CAD.

Response: Thank you for raising this point. Please see our reply to point #4. S1900A overexpression catalyzes *de novo* pyrimidine biosynthesis and deamidation to a lesser extent, thus enhancing virus production compared to the depletion of CAD.

6. Fig. S5B: there is still a reduction in NF κ B activity when CAD S1900A is overexpressed, the effect of CAD S1900A is similar to that of CAD WT, again that phosphorylation of CAD on S1900 is not the only mechanism how vCyc/cdk6 activate CAD.

Response: Thank you for the question. Please see our reply to comment #4. We think that S1900A overexpression compensated its weakened activity to deamidate RelA, thus leading to the reduction in NF- κ B activity. We agree with the reviewer that our data did not rule out the possibility that additional phosphorylation site(s) may be present in CAD that are mediated by vCyclin-CDK6. We have now discussed this in detail in the Discussion section (page 9, line 426-428).

7. Figure S7B: PALA does not inhibit BCBL-1 any better than BJAB, suggesting that its inhibition of CAD ATCase activity is not specific for KSHV-infected cells.

Response: The reviewer's point is well taken. We would like to point out that with limited numbers of cancer cells (in this case four cell lines), comparing drug sensitivities among them may not be sufficient to conclude whether CAD serves as a metabolic vulnerability for KSHV-infected cells. We thus tuned down our conclusions for Figure S7B (also Figures 7H, 7I) that inhibiting CAD blocked the cell viability of KSHV-positive lymphoma cells and a virus-negative Burkitt-like lymphoma cell line.

On the other hand, we did show that PALA preferentially reduced the cell viability for KSHV-infected TIME cells compared to mock-infected controls (Figure 6K). We have also performed new experiments using BJAB and BJAB-KSHV (freshly made in our lab) to show that both DON and PALA preferentially reduced the viability of KSHV-infected cells at lower concentration (new Figure 7J). We think that these data support our conclusions of CAD dependency for KSHV-infected cells.

Figure 6K

Figure 7J

8. Figure 7H: DON inhibits PEL cell lines as efficiently as (BC3) or better than (JSC1) BJAB, again suggesting that inhibiting CAD activity does not exert a preferential effect on KSHV-infected cells and KSHV-dependent proliferation. This would argue against one of the key arguments of this manuscript, i.e that the metabolic changes due to the activation of CAD by *vcyc/cdk6* could be exploited to specifically/preferentially inhibit the growth of KSHV-infected tumor cells.

Response: Thank you for the question. Please refer to our reply for comment #7. Furthermore, we found that specific virus-negative cancer cells are exploiting CAD via similar mechanisms (a dysregulated cellular cyclin D/CDK6-mediated CAD S1900 phosphorylation and activation). Thus, targeting CAD will not only be effective against KSHV-positive lymphomas, but will offer advantage to other cancers as well. We are now rigorously investigating the potential correlation between CAD hijacking and CAD reliance in multiple types of cancer with a library of more than 30 cell lines. We are confident that these analyses will provide us with a more definite answer, however, they are beyond the scope of the current study. We have provided additional discussion on CAD dependency for KSHV-positive and -negative cancers (page 9, line 402-408).

9. Figure S7 C-E: it would be important to show the effect of DON, PALA and Palbo on PEL cells in the colony forming assay in comparison to their effect on BJAB.

Response: Thank you for the suggestion. We have now tested DON, PALA (100 µM and 400 µM), and Palbociclib in PEL cells and BJAB (revised Figures S7E-S7G and new Figures S7H, S8A, and S8B). We found that (1) compared to virus-positive cell, BJAB showed resistance to DON, while Palbociclib efficiently blocked colony formation; (2) low concentration of PALA partially inhibited BC-3 cells but was not effective against BCBL-1, JSC-1, or BJAB; (3) high concentration of PALA inhibited the colony formation of BC-3 and BCBL-1 cells, while it showed partial effect on BJAB and JSC-1 cells. Similar to our response to comment #7, we wish to tune down our conclusion here that targeting CAD and CDK6 blocks colony formation for KSHV+/EBV+ lymphoma cells and BJAB.

Figure S7F

Figure S7G

Figure S7H

Figure S7I

Figure S8A

Figure S8B

Minor points:

1. Lines 178, 187, 197, 198, 229, 497: name of KSHV with a vcyc stop mutation not clear

Response: Thank you for the comment. We have corrected the name of the virus as ' Δ Cyclin' and kept it consistent throughout the manuscript.

2. Line 265, 267, 358: NFkB spelling incorrect

Response: Thank you for the comment. We have fixed the spelling errors.

3. Line 510: cells were spun, not 'spinned'

Response: We have fixed the spelling error.

Reviewer #3 (Remarks to the Author):

The submitted manuscript, "Hijacking of Nucleotide Biosynthesis and Deamidation-mediated Glycolysis by an Oncogenic Herpesvirus" by Wan et al., is a continuation of investigations focused on the role of pyrimidine synthesis, CAD deamidation activity, and glycolysis. The authors previously found that CAD deamidates the RelA subunit of NF- κ B to promote aerobic glycolysis and promote proliferation of cancer cells. Now they are applying that information to KSHV infection. The authors are proposing that KSHV vCyclin activates CDK6 to phosphorylate CAD leading to increased aerobic glycolysis and cell proliferation through deamidation of RelA. The study has several noteworthy observations (e.g., CAD phosphorylation at S1900 by CDK6 is activating, the necessity of CAD in KSHV tumorigenesis using an *in vivo* mouse model, and that vCyclin impacts metabolism in a CAD-dependent mechanism). The experiments use a variety of approaches and tools (several cells, *in vivo*, and *in vitro* assays). Some conclusions are drawn from well performed experiments, such as Fig 2I-J. Overall, the manuscript reports a novel, interesting concept; however, there are several opportunities for improvement and a need to support the claims that are made. Much of these comments are addressable by either inclusion of additional experiments or altering the manuscript to limit conclusions to what is possible with the current experiments.

Response: We thank the reviewer for the encouraging comments. The reviewer raised several important questions. Answers to these questions certainly will significantly advance our understanding on vCyclin-mediated metabolic reprogramming. We have strived to address the reviewer's questions with experiments to our best.

1. One of the major conclusions—i.e., that the effect of CAD on glycolysis in KSHV infection is a "RelA-deamidation-mediated glycolytic reprogramming" mechanism—is not directly shown and depends on evidence that is correlative. This conclusion appears to mostly be dependent on Fig 6H, which is not overly convincing by itself. It may be possible (likely?) that CAD is impacting glycolysis through a RelA-independent mechanism and that the role of RelA may be, at the most, a minor contributor.

Response: We appreciate the reviewer's point. To consolidate that vCyclin and CAD promotes glycolytic reprogramming in a RelA deamidation-dependent manner, we have depleted endogenous RelA with shRNA targeting its 3'UTR in HOK cells and then reconstituted them with RelA-WT or RelA-64A (the deamidation-resistant RelA). Overexpression of KSHV-vCyclin in HOK-RelA-WT cells induced the expression of glycolytic genes including *HK4* and *PDK3*, while such effect was abolished in HOK-RelA-64A cells (new Figure S2F). Our previous finding also showed that RelA-DD can fully restore the expression of the glycolytic genes upon CAD depletion (Figure S5B, *Cell Metabolism*, 2020). Based on these results, combined with our observation that KSHV-mediated glycolysis is RelA deamidation-dependent (Figure 6H, now 6I), we think that RelA deamidation is a key contributor to vCyclin- and CAD-mediated glycolysis reprogramming during KSHV infection.

Figure S2F

Cell Metabolism, 2020

2. The conclusions depend on many blots; however, no quantitative information is provided for the blots or through a complementing method.

Response: We appreciate the reviewer's suggestion. We have now added quantification of the blots where direct comparisons of the protein bands are needed (Figure 1F, 1G, S3E, S3F, and S3G). These data were repeated at least twice and representative results are shown. To complement the blots, we have also repeated the qPCR for the three metabolic genes in KSHV-infected HOK and TIME cells (new Figure 1D and 1E). For 2D blots showing RelA deamidation, we usually complement the results with qPCR data showing the upregulation of deamidation-dependent metabolic gene expression, and lactate assay/seahorse assay demonstrating the upregulation of glycolysis.

3. Moreover, it is not clear if the blots represent several independent biological replicates.

Response: Thank you for the excellent suggestion. We have added the information on the independent biological repeats for all blots in the respective figure legends.

4. Many claims regarding glycolysis and 'metabolism' are based only on lactate measurement, which may be insufficient to draw strong conclusions regarding the whole glycolytic pathway or a broader picture of the metabolic network.

Response: The reviewer's point is well taken. We would like to point out that lactate is the end product of glycolysis and reflects the rate of glycolysis (TeSlaa and Teitell, 2014). Lactate assay offers the advantage to be routinely used in regular lab settings with a plate reader and provides direct quantification. Meanwhile, we totally agree with the reviewer that monitoring the whole glycolytic

pathway is needed to consolidate our findings. In fact, we have performed Seahorse glycolysis rate assay (Figures 1J, S1C, S1D, and new Figure 2K) to monitor the real-time glycolytic changes during KSHV infection and vCyclin overexpression. We have also performed targeted metabolomics and isotope tracing experiments focusing on glycolytic intermediates including Glucose, F-6-P, F-1,6-BP, G-3-P, 1,3-BPG, PEP, DHAP, 3-PG, and Lactate (Figures 1B, 3B, 3C, and 3D). Due to technical limitations, we were not able to capture all intermediates in every experiment, but we have strived to get a broader picture of glycolysis.

Figure 2K

5. Similar to point #4, *de novo* pyrimidine synthesis is a major focus of conclusions; however, measurements of pyrimidines are rarely shown.

Response: The reviewer’s point is well taken. We would like to point out that many of the nucleotide intermediates and derivatives fall out of the detection limit of the mass spectrometry under our running condition. Therefore, a detailed analysis of pyrimidine biosynthesis requires extensive mass spectrometry experimentation. As we were interested in CAD regulation by vCyclin-CDK6, we focused on the reaction product of CAD dihydroorotic acid and its immediate downstream intermediate orotic acid (right diagram). We have tried to analyze carbamoyl phosphate (CAD reaction intermediate) but failed to detect it. The isotope tracing experiment was also purposefully done in 15 min to enrich the ¹⁵N-labeled CAD intermediates reflecting CAD activity. On the other hand, the incubation time limited us to measure downstream ¹⁵N-labeled pyrimidine intermediates.

To still address the reviewer’s concerns, we have now added new data showing ¹³C incorporation into uridine monophosphate (UMP), a key downstream pyrimidine intermediate, in WT- and ΔCyclin-infected HOK cells (new Figure S3J). From these results, combined with our ¹⁵N tracing results (Figures 3E, 3F, and 5C), we conclude that CAD activity is indeed promoted by phosphorylation in WT-infected cells, resulting in the increase of both upstream and downstream pyrimidine intermediates (right diagram).

Figure S3J

6. There does not appear to be a strong correlation between WT, S1900D, and S1900A regarding metabolism and virus replication. E.g., WT and S1900A have similar levels of dihydroorotic acid and lactate, but it appears that S1900A has a defect in virus replication as measured by PFU. This weakens the overall claim. Is there a reason for this observation?

Response: We appreciate the reviewer's insightful comment. Upon overexpression, we observed that S1900A and WT had similar levels of dihydroorotate (Figure 5C). We think that the majority of the WT CAD was not phosphorylated on S1900 upon overexpression in 293T cells, thus explaining the minimal difference between WT and S1900A. However in the context of KSHV infection, vCyclin will drive the phosphorylation and enzymatic activity of the WT protein but not the S1900A. Therefore, during KSHV infection WT behaved more like S1900D in promoting viral replication, whereas S1900A resulted in a lower PFU. We have added explanations in the Result section (page 6, line 274-275).

7. The manuscript lacks rationale, explanation or justification for the many different cells, time points, MOIs, metabolic genes examined, etc. reducing the ability of the work to be appreciated by a general audience.

Response: Thank you for the excellent suggestion. The different cell lines, time points, and MOIs were chosen and optimized based on published literatures (Choi et al., 2020; Dollery et al., 2021; Gong et al., 2014). We have now provided details in the Methods section. We have also added our rationale for monitoring the specific metabolic genes (HK4, PDK3, and PC) as a marker for RelA deamidation in the Results section (page 3, line 112-114).

8. The conclusion that "KSHV vCyclin promotes glycolysis and de novo pyrimidine synthesis, and is required for optimal viral replication" is not demonstrated by the experimental evidence. Does restoring glycolysis and de novo pyrimidine synthesis restore virus replication in the vCyclin mutant virus?

Response: We thank the reviewer for this insightful question. We have now performed *de novo* infection with KSHV- Δ Cyclin in uridine-supplemented HOKs. Uridine can be converted to UMP, UTP, CMP, etc. via the pyrimidine salvage pathway, thus compensating for the *de novo* pyrimidine biosynthesis. Addition of uridine significantly increased the viral titers of KSHV Δ Cyclin, though it failed to reach to a comparable level of KSHV-WT (new Figure 3I).

To induce deamidation-mediated glycolysis and viral gene expression, we depleted endogenous RelA via shRNA targeting its 3'UTR and reconstituted the HOKs with RelA-DD. We found that uridine addition and RelA-DD expression further restored the replication of Δ Cyclin (new Figure S3L), demonstrating the pivotal role of vCyclin in driving deamidation and pyrimidine biosynthesis. Notably, KSHV-WT still replicated better than Δ Cyclin in cells reconstituted with RelA-DD after uridine addition, while vCyclin complementation fully restored the metabolic gene expression and the replication of Δ Cyclin (new Figures S3H and S3M). This indicates that vCyclin has additional mechanisms to promote viral lytic replication, which is further discussed in the Discussion section (page 9, line 443-447).

Figure 3I

Figure S3L

Figure S3H

Figure S3M

9. Many experiments depend on genetic or small molecule/inhibitor treatment, however their effect on cell health/survivability is not included when interpreting the results. For example, infection with the vCyclin mutant virus causes a decrease in all the reported metabolites (Fig 3B). However, this may not be specific to glycolysis or pyrimidine synthesis as the authors conclude. Perhaps all metabolites are decreased due to differences in infection rate or the ability of the virus to replicate, or even the cell health, survivability.

Response: we found that knocking down CAD reduced the proliferation of HOK and 293T cells. Treating cells with DON or Palbo stopped cell proliferation, but no cell death was observed. Importantly, according to our own observations and also a previous publication (Gong et al., 2014), KSHV lytic replication will also stop cells from proliferation. Therefore, the total numbers of the live cells upon KSHV infection remain similar among the mock/treated and shCTL/shCAD groups (new Figures 6G and S7C). Therefore, we think that the cell health/survivability likely does not influence our interpretation of the results.

For Fig 3B, we have now included other metabolites belonging to different metabolic pathways including purine biosynthesis, NAD metabolism and TCA cycle. This showed that glycolysis intermediates, pentose phosphate pathway (PPP, a glycolysis branching pathway supporting nucleotide biosynthesis), and pyrimidines were clearly upregulated in WT-infected cells compared to Δ Cyclin. Purine intermediates (IMP, AMP) were also modestly upregulated. However, TCA cycle metabolites and NAD metabolism remained either unchanged or were reduced in WT virus infected cells compared to Δ Cyclin-infected cells. Thus, our data suggest that KSHV vCyclin reprograms specific metabolic pathways but does not elevate whole-cell metabolic activities. As our library of targeted metabolomics is limited, we were not able to get a broader picture of the metabolic network. There could be additional metabolic reprogramming activities governed by KSHV vCyclin which warrants further investigation. We have now added more discussion (page 9, line 443-447).

Figure 3B

Minor items to address:

1. Since the labeling data is provided for only a single 15 min time point, it is not possible to determine a rate and thus the usage of metabolic 'flux' and 'rate analysis' is not appropriate.

Response: Thank you for the comment. We have deleted 'flux' and 'rate analysis' and used alternative words ('determine the metabolic activities', 'intracellular labeled metabolites', and 'glycolysis tracing') to describe the tracing experiment (page 4, line 196-197, page 15, line 706-707).

2. Several data are presented as "relative" or "normalized" however it is not always clear what is meant by these terms.

Response: We have added the information regarding the 'relative' and 'normalized' in the methods. For qPCR results we usually presented as fold changes relative to the mock/uninfected control values. For mass spectrometry results the peak area (reflecting the intensity of the metabolite) is normalized by the cell number so we used 'normalized abundance'.

3. The image resolution of Fig 4E is too poor, the image is unreadable.

Response: We have now improved the resolution of Fig 4E. We have also expanded the size of it.

Reference:

1. Cai, Q., Lan, K., Verma, S.C., Si, H., Lin, D., and Robertson, E.S. (2006). Kaposi's sarcoma-associated herpesvirus latent protein LANA interacts with HIF-1 alpha to upregulate RTA expression during hypoxia: Latency control under low oxygen conditions. *Journal of virology* *80*, 7965-7975.
2. Choi, D., Park, E., Kim, K.E., Jung, E., Seong, Y.J., Zhao, L., Madhavan, S., Daghlian, G., Lee, H.H., Daghlian, P.T., *et al.* (2020). The Lymphatic Cell Environment Promotes Kaposi Sarcoma Development by Prox1-Enhanced Productive Lytic Replication of Kaposi Sarcoma Herpes Virus. *Cancer Res* *80*, 3130-3144.
3. Dollery, S.J., Maldonado, T.D., Brenner, E.A., and Berger, E.A. (2021). iTIME.219: An Immortalized KSHV Infected Endothelial Cell Line Inducible by a KSHV-Specific Stimulus to Transition From Latency to Lytic Replication and Infectious Virus Release. *Front Cell Infect Microbiol* *11*, 654396.
4. Gong, D., Wu, N.C., Xie, Y., Feng, J., Tong, L., Brulois, K.F., Luan, H., Du, Y., Jung, J.U., Wang, C.Y., *et al.* (2014). Kaposi's sarcoma-associated herpesvirus ORF18 and ORF30 are essential for late gene expression during lytic replication. *Journal of virology* *88*, 11369-11382.
5. Hollingworth, R., Stewart, G.S., and Grand, R.J. (2020). Productive herpesvirus lytic replication in primary effusion lymphoma cells requires S-phase entry. *J Gen Virol* *101*, 873-883.
6. Ma, T., Patel, H., Babapoor-Farrokhran, S., Franklin, R., Semenza, G.L., Sodhi, A., and Montaner, S. (2015). KSHV induces aerobic glycolysis and angiogenesis through HIF-1-dependent upregulation of pyruvate kinase 2 in Kaposi's sarcoma. *Angiogenesis* *18*, 477-488.
7. Manzano, M., Patil, A., Waldrop, A., Dave, S.S., Behdad, A., and Gottwein, E. (2018). Gene essentiality landscape and druggable oncogenic dependencies in herpesviral primary effusion lymphoma. *Nat Commun* *9*, 3263.
8. Tatibana, M., and Shigesada, K. (1972). Control of pyrimidine biosynthesis in mammalian tissues. V. Regulation of glutamine-dependent carbamyl phosphate synthetase: activation by 5-phosphoribosyl 1-pyrophosphate and inhibition by uridine triphosphate. *J Biochem* *72*, 549-560.

9. TeSlaa, T., and Teitell, M.A. (2014). Techniques to monitor glycolysis. *Methods Enzymol* 542, 91-114.
10. Wu, Y., Shrestha, P., Heape, N.M., and Yarchoan, R. (2022). CDK4/6 inhibitors sensitize gammaherpesvirus-infected tumor cells to T-cell killing by enhancing expression of immune surface molecules. *J Transl Med* 20, 217.
11. Yogev, O., Lagos, D., Enver, T., and Boshoff, C. (2014). Kaposi's sarcoma herpesvirus microRNAs induce metabolic transformation of infected cells. *PLoS Pathog* 10, e1004400.

REVIEWER COMMENTS

Reviewer #1 (Remarks to the Author):

The revised manuscript by Wan et al., has addressed the concerns raised during the initial reviews. The experimental evidence provided is very convincing to support the conclusions in the manuscript. Also, the article is rewritten to considerably higher clarity. I have no further comments.

Reviewer #2 (Remarks to the Author):

In this very interesting manuscript, Wan and colleagues report that the KSHV viral cyclin homologue activates the CAD multifunctional enzyme by mediating, in complex with cellular cdk6, its phosphorylation on S1900, and thereby activating its ability to deamidate RelA and stimulate CAD-dependent metabolic changes in the KSHV-infected cell. This is a very interesting observation which could potentially be exploited to devise new therapies for KSHV-associated tumors. In fact, the authors provide some experimental evidence that available inhibitors that interfere with the pathways identified in this manuscript could be repurposed to treat primary effusion lymphoma.

The authors have carefully addressed most of the reviewers comments. I found two point that might still merit attention:

1. Line 346-349, Figure 7J: "With mock-infected and KSHV-infected BJAB cells, we found that DON and PALA preferentially inhibited cell proliferation for BJAB-KSHV at lower doses (Figure 7J), suggesting that CAD may serve as a metabolic vulnerability for KSHV-infected cells."

In Figure 7J, the IC50 values for DON and PALA appear to be the same for KSHV-infected and uninfected BJAB. I am therefore not sure that one can conclude that "DON and PALA preferentially inhibited proliferation of BJAB-KSHV at lower doses".

2. Line 96: „viral homology“ should be „viral homologue“

Reviewer #3 (Remarks to the Author):

Wan and colleagues provided an extensive response to all three of the previous reviewers' comments. It is noteworthy that they included additional experimental evidence to support their study and respond to concerns raised by the reviewers. The authors should be commended for their hard work and for toning down some of the previous conclusions. The manuscript is much improved.

This reviewer would recommend some additional editing to the Discussion. Most significantly, the authors should include a discussion regarding the indirect versus direct effects raised by Reviewer #2 (comments 2-3). The authors can make their case that it is a direct effect and leave it up to readers to weigh the evidence and that authors claim. Similarly, the authors should include a discussion regarding reviewer #3 comment 9 (impacts due to alteration in virus replication or host health) and reviewer #1 comment 3 (differences observed in the metabolic pathway). Again, the authors can make their case and leave it up to the reader to judge.

The authors should discuss potential differences in media composition among their experiments. It appears that most experiments were done in DMEM without pyruvate but the Seahorse measurements were done in 1mM pyruvate. This is important since the addition of pyruvate can later lactate production and glycolytic activity. As related note, ECAR measurements on the Seahorse are largely driven by lactate production, the dependence of a conclusion on lactate measurements when supported by ECAR measurement are not independently confirmatory.

General Response:

We thank all three reviewers for their constructive comments on our manuscript. We have revised our conclusions in the manuscript and provided more discussions based on the comments raised by reviewer #2 and reviewer #3. We hope that the reviewers will appreciate the improvement of our revised manuscript.

REVIEWER COMMENTS

Reviewer #1 (Remarks to the Author):

The revised manuscript by Wan et al., has addressed the concerns raised during the initial reviews. The experimental evidence provided is very convincing to support the conclusions in the manuscript. Also, the article is rewritten to considerably higher clarity. I have no further comments.

We thank the reviewer for the appreciation of our revision.

Reviewer #2 (Remarks to the Author):

In this very interesting manuscript, Wan and colleagues report that the KSHV viral cyclin homologue activates the CAD multifunctional enzyme by mediating, in complex with cellular cdk6, its phosphorylation on S1900, and thereby activating its ability to deamidate RelA and stimulate CAD-dependent metabolic changes in the KSHV-infected cell. This is a very interesting observation which could potentially be exploited to devise new therapies for KSHV-associated tumors. In fact, the authors provide some experimental evidence that available inhibitors that interfere with the pathways identified in this manuscript could be repurposed to treat primary effusion lymphoma.

We are thrilled to see that the reviewer appreciated the significance of our study. We have made the according changes in our manuscript based on the comments.

The authors have carefully addressed most of the reviewers' comments. I found two point that might still merit attention:

1. Line 346-349, Figure 7J: "With mock-infected and KSHV-infected BJAB cells, we found that DON and PALA preferentially inhibited cell proliferation for BJAB-KSHV at lower doses (Figure 7J), suggesting that CAD may serve as a metabolic vulnerability for KSHV-infected cells."

In Figure 7J, the IC50 values for DON and PALA appear to be the same for KSHV-infected and uninfected BJAB. I am therefore not sure that one can conclude that "DON and PALA preferentially inhibited proliferation of BJAB-KSHV at lower doses".

Thank you for the comments. Indeed, the IC50 of DON remained similar for BJAB and BJAB-KSHV even though both inhibitors had a better effect in inhibiting BJAB-KSHV at lower doses. We have thus toned down the conclusion for this experiment in the manuscript (line 347). As BJAB is a cancer cell line, we think that it is perhaps not the best cell model to reflect the metabolic reprogramming and metabolic vulnerability induced by KSHV (as compared to Figure 6K for TIME and TIME-KSHV cells).

2. Line 96: „viral homology“ should be „viral homologue”

Thank you for your suggestion. We have fixed it in the manuscript.

Reviewer #3 (Remarks to the Author):

Wan and colleagues provided an extensive response to all three of the previous reviewers' comments. It is noteworthy that they included additional experimental evidence to support their study and respond to concerns raised by the reviewers. The authors should be commended for their hard work and for toning down some of the previous conclusions. The manuscript is much improved.

We thank the reviewer for the appreciation of the significant improvement of our revision. We have now provided more discussions based on the comments.

This reviewer would recommend some additional editing to the Discussion. Most significantly, the authors should include a discussion regarding the indirect versus direct effects raised by Reviewer #2 (comments 2-3). The authors can make their case that it is a direct effect and leave it up to readers to weigh the evidence and that authors claim. Similarly, the authors should include a discussion regarding reviewer #3 comment 9 (impacts due to alteration in virus replication or host health) and reviewer #1 comment 3 (differences observed in the metabolic pathway). Again, the authors can make their case and leave it up to the reader to judge.

We thank the reviewer for the suggestion. We have included the discussion on the direct and indirect effects of CDK6 for KSHV-mediated metabolic reprogramming in our manuscript (Line 454). We have also provided discussions for reviewer #3 comment 9 (Line 456) and reviewer #1 comment 3 (Line 429).

The authors should discuss potential differences in media composition among their experiments. It appears that most experiments were done in DMEM without pyruvate but the Seahorse measurements were done in 1mM pyruvate. This is important since the addition of pyruvate can later lactate production and glycolytic activity. As related note, ECAR measurements on the Seahorse are largely driven by lactate production, the dependence of a conclusion on lactate measurements when supported by ECAR measurement are not independently confirmatory.

The reviewer's point is well-taken. We reached out to the Seahorse technical service. Based on their feedback, the 1 mM pyruvate included in the Seahorse assay medium is to maintain similar composition with cell culture medium and avoid inducing cellular stress during the assay due to changes in media composition. The concentration of pyruvate in our growth media is as follows: DMEM (Hyclone): 1 mM, KGM-2: 0.5 mM, Vasculife endothelial medium: 1.04 mM.

The Seahorse team thinks that the extracellular pyruvate is mostly taken into the mitochondrial instead of forming lactate (due to low NADH in the cytosol). They have validated in 20 cell lines that incubation in assay media containing pyruvate but without glucose resulted in the failure to detect significant formation of lactate in the extracellular media.

On the other hand, we agree with the reviewer that the media composition is not the same for the Seahorse media as the growth media. Thus, one should expect the amount of lactate produced is different between the lactate assay and the Seahorse assay. However, both assays demonstrated (though not independently) the upregulation of glycolysis (lactate) for KSHV-infected and vCyclin-transduced cells, supporting our hypothesis that vCyclin contributes to KSHV-driven glycolysis. New discussion is now included (Line 383).